

# Polarization lidar: An extended three-signal calibration approach

Cristofer Jimenez[1], Albert Ansmann[1], Ronny Engelmann[1], Moritz Haarig[1], Jörg Schmidt[2], and Ulla Wandinger[1]

[1]Leibniz Institute for Tropospheric Research, Leipzig, D-04318, Germany
[2]University of Leipzig, Institute for Meteorology, Leipzig, D-04103, Germany

*Correspondence to*: Cristofer Jimenez (cristofer.jimenez@tropos.de)

**Abstract.** We present a new formalism to calibrate a three-signal polarization lidar and to measure highly accurate height profiles of the volume linear depolarization ratios under realistic experimental conditions. The methodology considers elliptically polarized laser light, angular misalignment of the receiver unit with respect to the main polarization plane of the laser pulses, and cross-talk between the receiver channels. A case study of a liquid-water cloud observation demonstrates the potential of the new technique. Long-term observations of the calibration parameters corroborate the robustness of the method and the long-term stability of the three-signal polarization lidar. A comparison with another polarization lidar shows excellent agreement regarding the derived volume linear polarization ratio of biomass burning smoke throughout the troposphere and the lower stratosphere up to 16 km height.

## 1 Introduction

Atmospheric aerosol particles influence the evolution of clouds and the formation of precipitation in complex and not well understood ways. Strong efforts are needed to improve our knowledge about aerosol-cloud interaction and the parameterization of cloud processes in atmospheric (weather and climate) models, weather forecasts, and especially to decrease the large uncertainties in future climate predictions (IPCC 2013). Besides more measurements in contrasting environments with different climatic and air pollution conditions, new experimental (profiling) methods need to be developed to allow an improved and more direct observation of the impact of different aerosol types and mixtures on the evolution of liquid-water, mixed-phase, and ice clouds occurring in the height range from the upper planetary boundary layer to the tropopause. Active remote sensing is a powerful technique to continuously and coherently monitor the evolution and life cycle of clouds in their natural environment.

Recently, Schmidt et al. (2013, 2014, 2015) introduced the so-called dual-field-of-view (dual-FOV) Raman lidar technique which allows us to measure aerosol particle extinction coefficients (used as aerosol proxy) close to cloud base of a liquid-water cloud layer and to retrieve, at the same time, cloud microphysical properties such as cloud droplet effective radius and cloud droplet number concentration (CDNC) in the lower part of the cloud layer. In this way, the most direct impact of aerosol particles on cloud microphysical properties could be determined. However, the method is only applicable after sun set (during nighttime) and signal averaging of the order of 10-30 minutes is required to reduce the impact of signal noise on the observations to a tolerable level. As a consequence, cloud properties cannot be resolved on scales of 100-200 m horizontal resolution or 10-30 s. To improve the dual-FOV measurement concept towards daytime observations and shorter signal



averaging times (towards time scales allowing us to resolve individual, single updrafts and downdrafts) we developed the so-called dual-FOV polarization lidar method (Jimenez et al., 2017, 2018a). This technique makes use of strong depolarization of transmitted linearly polarized laser pulses in water clouds by multiple scattering of laser photons by water droplets (with typical number concentrations of 100 cm$^{-3}$). This novel polarization lidar method can be applied to daytime observations with

resolutions of 10-30 s. An extended description of the method is in preparation (Jimenez et al., 2018b).

Highly accurate observations of the volume linear depolarization ratio are of fundamental importance for a successful retrieval of cloud microphysical properties by means of the new polarization lidar technique. In this article (part 1 of a series of several papers on the dual-FOV polarization lidar technique), we present and discuss our new polarization lidar setup and how the lidar channels are calibrated. The basic product of a polarization lidar is the volume linear depolarization ratio, defined as the

ratio of the cross-polarized to the co-polarized atmospheric backscatter intensity, and is derived from lidar observations of the cross and co-polarized signal components, or alternatively, from the observation of the cross-polarized and total (cross + co-polarized) signal components. Cross and co-polarized denote the plane of linear polarization, orthogonal and parallel to the linear polarization plane of the transmitted laser light, respectively. Reichardt et al. (2003) proposed a robust concept to obtain high-quality depolarization ratio profiles by measuring simultaneously three signal components, namely the cross and co-

polarized signal components and additionally the total elastic backscatter signal. We will follow this idea as described in Sect. 2. Reichardt et al. (2003) assumed that the laser pulses are totally linearly polarized. Recent studies, however, have shown that the transmitted laser pulses can be slightly elliptically polarized (David et al., 2012; Freudenthaler, 2016; Bravo-Aranda et al., 2016; Belegante et al., 2018). We will consider this effect in our extended approach of the three-channel depolarization technique. We further extend the formalism by considering realistic strengths of cross talk between the three channels and we

propose a practical inversion scheme based on the determination of the instrumental constants for the retrieval of high temporal resolution volume depolarization ratio profiles.

The article is organized as follows. In section 2, the lidar instrument is described. The new methodology to calibrate the lidar system and to obtain high quality depolarization ratio observations is outlined in Sect. 3. Section 4 presents and discusses atmospheric measurements performed to check and test the applicability of the new methodology. Concluding remarks are

given in Sect. 5.

## 2 Lidar setup

A sketch of the instrumental setup, providing an overview of the entire lidar system, is shown in Fig. 1. MARTHA (Multiwavelength Tropospheric Raman lidar for Temperature, Humidity, and Aerosol profiling) has a powerful laser

transmitting in total 1~J per pulse at a repetition rate of 30~Hz and has an 80~cm telescope, and is thus well designed for tropospheric and stratospheric aerosol observations (Mattis et al., 2004, 2008, 2010; Schmidt et al., 2013, 2014, 2015; Jimenez et al., 2017, 2018). MARTHA belongs to the European Aerosol Research Lidar Network (EARLINET) (Pappalardo et al., 2014). We implemented a new three-signal polarization lidar receiver unit to the left side of the large telescope (see Fig. 1). The new receiver setup is composed of three independent telescopes co-aligned with the lidar transmitter.



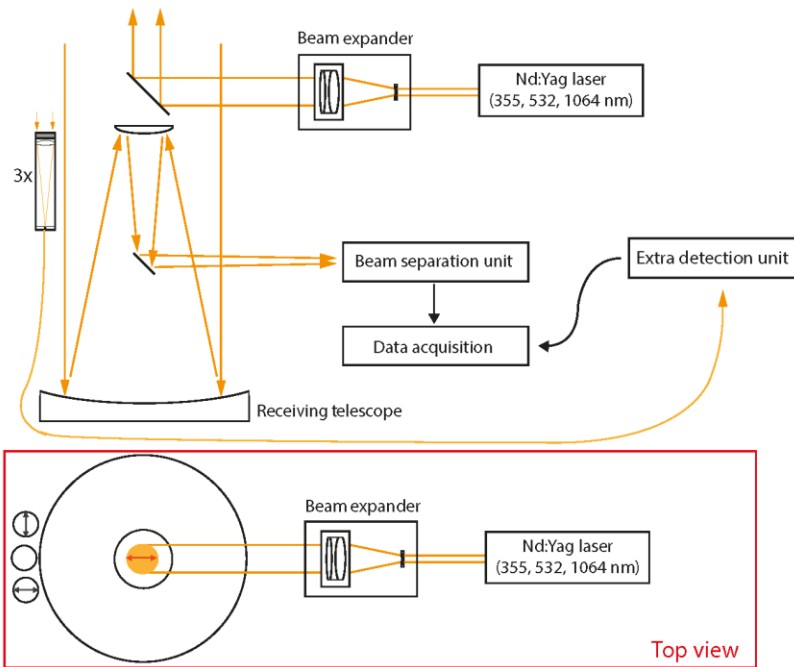

**Figure 1: Overview of the EARLINET lidar MARTHA. The three-signal receiver unit of the new polarization lidar setup (details are shown in Fig. 2) is integrated into the MARTHA telescope construction (left side in both of the two sketches). The outgoing laser beam is 54 cm away from the new polarization-sensitive receiver unit. The main plane of linear polarization of the laser pulses and the polarization sensitivity of cross- and co-polarized receiver channels are indicated by arrows in the top-view sketch.**

Figure 2 provides details of the new polarization-sensitive channels. Each of the small receiver telescopes consist of 2" achromatic lens with a focal length of 250 mm. An optical fiber with an aperture of 400 µm is placed at the focal point of the lens. The resulting field of view (FOV) is 1.6 mrad. The receivers have in principle the same overlap function, since they are identical and are implemented into the large telescope at the same distance from the laser beam axis. The laser-beam receiver-FOV overlap is complete at about 650 m above the lidar.

At the output of the fiber a 2 mm ball lens is placed (scrambler in Fig. 2) in order to remove the small sensitivity of the interference filter to the changing incidence angle of backscattered light in the near-range. Only above 650 m (full overlap), we can assume that all light from all heights is backscattered at exactly 180°. A spatial attenuation unit which consists of two optical fibers is integrated in the receiver setup, replacing the usual setup with neutral density filters. The distance between the two fibers with given aperture can be changed and thus the strength of the incoming lidar return signal. The attenuation factor depends on the square of the distance between the fibers and on the numerical aperture of the fibers. E.g., signal attenuation by a factor of about 100 when the distance is 25 mm, and about 1000 with 79 mm distance.





The purpose of the new receiver system is to measure accurate profiles of the volume depolarization ratio in clouds between 1 and 12 km height. For the separation of the polarization components two of the three polarization telescopes are equipped with a linear polarization filter (see Fig. 2, linear polarizer) in front of the entrance lens. In the alignment process, the cross-polarized axis is found when the count rates are at the minimum. The co-polarized channel is then rotated by 90° compared to the cross-

5    polarized filter position, because it is set manually, the difference between the true polarization axis of the filters may not be 90°, however, in this approach we will assume it, since the impact of small variations in the pointing angles of the polarization filters can be neglected (see Appendix A). Additionally, a small tilt between the finally obtained polarization plane of the receiver unit and the true polarization state (main plane of linear polarization) of the transmitted laser pulses is expected and thus assumed in the methodology outlined in Sect. 3.

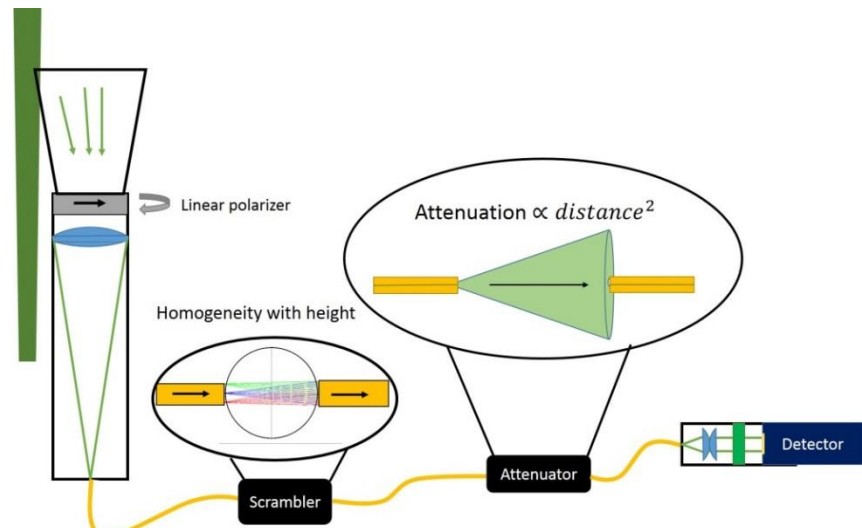

**Figure 2: Sketch of one of the three identical receiver channel of the three-signal polarization lidar. The different parts are explained in the text.**

15   **3 Methodology**

In Sect. 3.1, we begin with definitions and equations that allow us to describe the transmission of polarized laser pulses into the atmosphere, backscatter, extinction, and depolarization of polarized laser radiation by the atmospheric constituents, and the influence of the receiver set up on the depolarization ratio measurements. Based on this theoretical framework we will derive three lidar equations for our three measured signal components. In Sect. 3.2, we then present the derivation of the new

20   three-signal method for the determination of the volume depolarization ratio starting from the three lidar equations (one for each channel) defined in Sect. 3.1.



### 3.1 Theoretical background: Three-signal polarization lidar

We follow the notation and explanations of Freudenthaler (2016), Bravo-Aranda et al. (2016), and Belegante et al. (2018) in the description of the lidar setup, from the laser source (as part of the transmitter unit) to the detector unit (as part of the receiver block), and regarding the interaction of the polarized laser light photons with atmospheric particles and molecules by means of the Müller- Stokes formalism (Chipman, 2009). A Stokes vector describes the flux and the state of polarization of the transmitted laser radiation pulses and Müller matrices describe how the optical elements of the transmitter and receiver units and the atmospheric constituents change the Stokes vector. The laser beam is expanded before transmission into the atmosphere. In most polarization lidar applications it is assumed that the transmitted laser radiation is totally linearly polarized. But this is not the case in practice. In our approach, we therefore take into consideration that the transmitted wave front contains a non-negligible, small amount of cross-polarized light after passing through the beam expander. Additionally, we consider a small-angular misalignment, described by angle $\theta$ between the main plane of polarization of the laser beam and the orientation of the respective plane of polarization defined by the polarization filters in front of the telescopes of the receiver unit of our three-channel polarization lidar configuration described below.

The transmitted radiation $P_0(z)$ of the laser pulse can be written as the sum

$$P_0 = P_{0,II} + P_{0,\perp} \tag{1}$$

with the co- and cross-polarized light components, $P_{0,II}$ and $P_{0,\perp}$, with polarizations parallel and orthogonal to the main plane of laser light polarization. We introduce the so-called cross-talk term $\varepsilon_l$,

$$\varepsilon_l = \frac{P_{0,\perp}}{P_{0,II}}, \tag{2}$$

which describes the small amount of cross-polarized light in the laser beam after leaving the transmission block of the lidar towards the atmosphere. Now we can write:

$$P_0 = (1 + \varepsilon_l)P_{0,II}. \tag{3}$$

The transmitted electromagnetic wave front is then given by the Stokes vector (Lu and Chipman, 2009)

$$\boldsymbol{I_L} = P_{0,II}\begin{pmatrix} 1+\varepsilon_l \\ 1-\varepsilon_l \\ 0 \\ 0 \end{pmatrix} = P_0\begin{pmatrix} 1 \\ \frac{1-\varepsilon_l}{1+\varepsilon_l} \\ 0 \\ 0 \end{pmatrix}. \tag{4}$$

The misalignment between the polarization axis of the transmitted light and the co-polarized receiver channel (defined by the respective polarization filter in front of the PMT) is characterized by angle $\theta$ and considered by the rotation Müller matrix (Bravo-Aranda et al., 2016):

$$\boldsymbol{R(\theta)} = \begin{pmatrix} 1 & 0 & 0 & 0 \\ 0 & \cos(2\theta) & -\sin(2\theta) & 0 \\ 0 & \sin(2\theta) & \cos(2\theta) & 0 \\ 0 & 0 & 0 & 1 \end{pmatrix}. \tag{5}$$





Then the incident field after backscattering by atmospheric particles and molecules, and before passing the receiver block can be written as (Freudenthaler, 2016):

$$\boldsymbol{I_{in}} = \mathbf{FR(\theta)}\boldsymbol{I_L} = F_{11}\begin{pmatrix} 1 & 0 & 0 & 0 \\ 0 & a & 0 & 0 \\ 0 & 0 & -a & 0 \\ 0 & 0 & 0 & 1-2a \end{pmatrix}\begin{pmatrix} 1 & 0 & 0 & 0 \\ 0 & \cos(2\theta) & -\sin(2\theta) & 0 \\ 0 & \sin(2\theta) & \cos(2\theta) & 0 \\ 0 & 0 & 0 & 1 \end{pmatrix}P_0\begin{pmatrix} 1 \\ \frac{1-\varepsilon_l}{1+\varepsilon_l} \\ 0 \\ 0 \end{pmatrix},$$

$$\boldsymbol{I_{in}} = F_{11}P_0\begin{pmatrix} 1 \\ \frac{1-\varepsilon_l}{1+\varepsilon_l}\cos(2\theta)a \\ -\frac{(1-\varepsilon_l)}{1+\varepsilon_l}\sin(2\theta)a \\ 0 \end{pmatrix} \qquad (6)$$

with the atmospheric polarization parameter

$$a = \frac{1-\delta}{1+\delta}. \qquad (7)$$

The scattering matrix $\mathbf{F}$ describes the interaction of the laser photons with the atmospheric particles and molecules. $F_{11}$ and $\delta$ are the backscatter coefficient and the volume linear depolarization ratio, respectively.

The true volume backscatter coefficient ($\beta := F_{11}$) is given by

$$\beta = \beta_{II} + \beta_{\perp} = (1+\delta)\beta_{II} \qquad (8)$$

with the backscatter contributions for the co- and cross-polarization planes (with respect to the true polarization planes given by the transmitted laser pulses). The volume linear depolarization ratio is defined as

$$\delta(z) = \frac{\beta_{\perp}(z)}{\beta_{II}(z)}. \qquad (9)$$

Figure 3 illustrates the different polarization states and configurations of the original laser pulses (Fig. 3a) and after leaving the beam expander as elliptically polarized laser light (Fig. 3b). The receiver block may be not well aligned to the main plain of laser radiation so that the PMT measures different cross- and co polarized signal components with respect the outgoing cross- and co-polarized laser light components in Fig. 3b. The rotated polarization axis is represented in Fig. 3c, and after being backscattered and depolarized, the incident polarization plane has the form as shown in Fig. 3d.

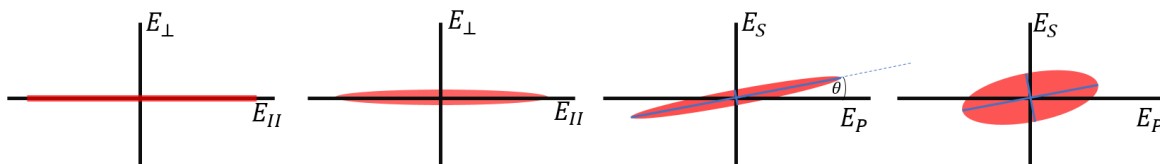

**Figure 3: (a) Polarization state of the light generated by the laser (100% linearly polarized), $E$ denotes electromagnetic field. (b) The laser radiation is elliptically polarized after passing the beam expander (see Fig. 1). (c) The receiving cross- and co-polarized signal channels ($E_S$ and $E_P$) are usually not perfectly aligned to the main polarization plane of the laser radiation, i.e. $\theta > 0$. (d) Polarization plane in the receiver for light which has been backscattered and depolarized by the atmosphere.**



To distinguish the apparent, measured volume backscatter coefficient, determined from the actually measured co- and cross-polarized signal components which are related to the incident field $\boldsymbol{I}_{in}$ (Eq. (6), see Fig. 3c) we introduce index 'in' and have the following relationships and links to the (true) laser light polarization plane:

$$\beta_{in} = \beta_{II,in} + \beta_{\perp,in} = \beta, \tag{10}$$

$$\beta_{II,in} - \beta_{\perp,in} = \frac{1-\varepsilon_l}{1+\varepsilon_l}\cos(2\theta)\, a\beta. \tag{11}$$

Using now Eq. (10) (describing the first term of $\boldsymbol{I}_{in}$ in Eq. (6)) and Eq. (11) (describing the second term of $\boldsymbol{I}_{in}$ in Eq. (6)), the apparent backscatter components $\beta_{II,in}$ and $\beta_{\perp,in}$ can be written as:

$$\beta_{II,in} = \left(1 + \frac{(1-\delta)}{(1+\delta)}\frac{(1-\varepsilon_l)}{(1+\varepsilon_l)}\cos(2\theta)\right)\beta/2, \tag{12}$$

$$\beta_{\perp,in} = \left(1 - \frac{(1-\delta)}{(1+\delta)}\frac{(1-\varepsilon_l)}{(1+\varepsilon_l)}\cos(2\theta)\right)\beta/2. \tag{13}$$

These three backscattering components (Eqs. (10), (12), and (13)) can be measured separately using the three different telescopes of our polarization lidar described in Sect. 2.

It is worthwhile to mention that polarization lidars typically have two detection channels, either a cross-polarized and a parallel-polarized channel or a cross-polarized and so-called total channel. A commonly used method for the calibration is the to insert an extra polarization filter into the optical path of the receiver unit and to rotate or tilt a $\lambda/2$ plate (Liu and Wang, 2013; Engelmann et al., 2016, McCullough et al., 2017). For these calibrations an extra measurement period is required. This calibration can introduce new and significant uncertainties (Biele et al., 2000; Freudenthaler et al., 2009; Mattis et al., 2009; Haarig et al., 2017).

As mentioned in the introduction, the concept to calibrate a lidar depolarization receiver by using three channels was proposed by Reichardt et al. 2003. The method consists of an absolute calibration procedure based on the measurement of elastically backscattered light with three detection channels for measuring co-, cross- and totally polarized backscatter components.

To determine the number of counts that the detection channels measure, Müller Matrices representing the optical path of each channel would need to be added on Eq. (6). Nevertheless, in this approach we follow the view adopted by Reichardt et al. (2003), where the traditional lidar equation is used to characterize the lidar channels.

Let us now introduce the lidar equations for these three signals. Following Reichardt et al. (2003), the number of photons $N_i$ that a lidar detects at height $z$ (above the full overlap height) with channel $i$ is given by

$$N_i(z) = P_0\left(\eta_{II,i}\beta_{II,in}(z) + \eta_{\perp,i}\beta_{\perp,in}(z)\right)T^2(z)/z^2. \tag{14}$$

$P_0$ is the emitted number of emitted laser photons and $\eta_{II,i}$ and $\eta_{\perp,i}$ are the optical efficiencies regarding the co- and cross-polarized components ($\beta_{II,in}$ and $\beta_{\perp,in}$) of the backscattered light that arrives at the channel-$i$ detector. These efficiencies include instrumental constants that contain the total transmittance through all optical components and gain of the detectors and attenuation in the path of each channel. $T$ denotes the atmospheric single-path transmission and is the same for all three detection channels (co, cross and total), since the extinction is independent of the state of polarization of the light. Rearrangements lead to the following versions of the lidar equations for the cross ($S$) and co-polarized ($P$) channels:





$$N_i(z) = P_0\, \eta_{II,i}\left(\beta_{II,in}(z) + D_i\beta_{\perp,in}(z)\right)T^2(z)/z^2, \tag{15}$$

or

$$N_i(z) = P_0\, \eta_{\perp,i}\left(D_i^{-1}\beta_{II,in}(z) + \beta_{\perp,in}(z)\right)T^2(z)/z^2, \tag{16}$$

here $D_i$ denotes the so-called efficiency ratio (Reichardt et al., 2003), and it is defined as:

$$D_i := \frac{\eta_{\perp,i}}{\eta_{II,i}}, \tag{17}$$

Because identical polarization filters are used in our lidar setup, we can assume $D_p = D_S^{-1}$. In the case of the total signal component ($i=$ tot) we introduce the overall efficiency $\eta_{tot}$ for simplicity reasons. The numbers of photons measured with each of the three channels ($i$ = P, S, tot) are then given by

$$N_P(z) = P_0\eta_{II,P}\left(\beta_{II,in}(z) + D_P\beta_{\perp,in}(z)\right)T^2(z)/z^2, \tag{18}$$

$$N_S(z) = P_0\eta_{\perp,S}\left(\beta_{\perp,in}(z) + D_S^{-1}\beta_{II,in}(z)\right)T^2(z)/z^2, \tag{19}$$

$$N_{\text{tot}}(z) = P_0\eta_{tot}\beta_{in}(z)T^2(z)/z^2. \tag{20}$$

After further rearranging we finally obtain:

$$\frac{N_P(z)z^2}{\eta_{II,P}P_0 T^2(z)} = \beta_{II,in}(z) + D_P\beta_{\perp,in}(z), \tag{21}$$

$$\frac{N_S(z)z^2}{\eta_{\perp,S}P_0 T^2(z)} = \beta_{\perp,in}(z) + D_S^{-1}\beta_{II,in}(z), \tag{22}$$

$$\frac{N_{tot}(z)z^2}{\eta_{tot}P_0 T^2(z)} = \beta_{in}(z). \tag{23}$$

To consider, in the next step, receiver misalignment and cross talk effects, we introduced the parameters $\varepsilon_l = \frac{P_{0,\perp}}{P_{0,II}}$ (Eq. (2)), describing the small amount of cross-polarized light in the laser beam after leaving the transmission block into the atmosphere, and the rotation angle $\theta$, describing the angular misalignment between the transmitter and receiver units. To consider also the receiver-channel cross talk, we further introduce $\varepsilon_r$, defined by $\varepsilon_r = D_S^{-1} = D_P$. The receiver cross talk value is typically $\varepsilon_r \le 10^{-3}$ (according to the filter manufacturer) as here the only element to consider is the polarization filter in front of the telescopes. Combining now Eqs. (10), (12), and (13) with Eqs. (21) - (23), we can write:

$$\frac{N_P(z)z^2}{\eta_{II,P}\, P_0 T^2(z)} = \beta_{II,in}(z) + \varepsilon_r\beta_{\perp,in}(z) = \left(1 + \varepsilon_r + \frac{(1-\delta(z))}{(1+\delta(z))}\frac{(1-\varepsilon_l)}{(1+\varepsilon_l)}(1 - \varepsilon_r)\cos(2\theta)\right)\beta(z)/2, \tag{24}$$

$$\frac{N_S(z)z^2}{\eta_{\perp,S}P_0 T^2(z)} = \beta_{\perp,in}(z) + \varepsilon_r\, \beta_{II,in}(z) = \left(1 + \varepsilon_r - \frac{(1-\delta(z))}{(1+\delta(z))}\frac{(1-\varepsilon_l)}{(1+\varepsilon_l)}(1 - \varepsilon_r)\cos(2\theta)\right)\beta(z)/2, \tag{25}$$

$$\frac{N_{tot}(z)z^2}{\eta_{tot}P_0 T^2(z)} = \beta_{in}(z) = \beta(z). \tag{26}$$

Until this point, the analytical procedure has been based on the assumption that the polarization filters in front of the cross- and co-polarized telescopes are pointing 90° with respect to each other. However, in the general case, when their angular deviation with respect to their respective components is different ($E_P$ to $E_{II}$ and $E_S$ to $E_\perp$), Eqs. (24) and (25) have a different angular component. In this approach, we keep this assumption for the development of a simple calibration procedure. In





Appendix A, the general case is evaluated (angle P to S ≠ 90°), and based on a measurement example, we demonstrated that the impact of this assumption can be neglected in our system.

**3.2 Determination of calibration constants and the volume linear depolarization ratio**

5 Outgoing from Eqs. (24)-(26) we will define instrumental (inter-channel) constants which are required to calibrate the lidar in the experimental practice and which are also used in the determination of the volume linear depolarization ratio. The equations for the determination of the depolarization ratios will be given. Three different ways can be used to determine the linear depolarization ratio profiles.

Considering Eq. (26) and the sum of Eqs. (24) and (25), we can write

$$\frac{N_{tot}(z)}{\eta_{tot}} = \frac{1}{1+\varepsilon_r}\left(\frac{N_P(z)}{\eta_{II,P}} + \frac{N_S(z)}{\eta_{\perp,S}}\right), \tag{27}$$

Eq. (27) is independent of the transmission cross talk factor $\varepsilon_l$ and of the rotation of the receiver axis (and thus rotation angle $\theta$), but dependents of the receiver cross talk factor $\varepsilon_r$.

Let us introduce the following inter-channel instrumental constants

$$X_P = \frac{\eta_{tot}}{(1+\varepsilon_r)\eta_{II,P}}, \tag{28}$$

$$X_S = \frac{\eta_{tot}}{(1+\varepsilon_r)\eta_{\perp,S}}, \tag{29}$$

$$X_\delta = \frac{\eta_{II,P}}{\eta_{\perp,S}} = \frac{X_S}{X_P} \tag{30}$$

and the signal ratios $R_P, R_S, R_\delta$

$$R_P(z) = \frac{N_P(z)}{N_{tot}(z)}, \tag{31}$$

$$R_S(z) = \frac{N_S(z)}{N_{tot}(z)}, \tag{32}$$

$$R_\delta(z) = \frac{N_S(z)}{N_P(z)}. \tag{33}$$

By using these definitions, Eq. (27) (after multiplication with $\frac{\eta_{tot}}{N_{tot}(z)}$) can be rearranged to

$$X_P R_P(z) + X_S R_S(z) = 1. \tag{34}$$

Eq. (34) is only valid for the case of an almost ideal polarization lidar receiver unit, i.e., when $D_S^{-1} = D_P$ ($= \varepsilon_r$). This is not the case for most of lidar systems where the receiver and separation unit may introduce differences between the transmission

25 ratios $D_S^{-1}$ and $D_P$. In the next step, we form the difference of Eq. (34) for altitude $z_j$ minus Eq. (34) for altitude $z_k$ and obtain:

$$X_\delta(z_j, z_k, t) = -\frac{R_P(z_j,t)-R_P(z_k,t)}{R_S(z_j,t)-R_S(z_k,t)}, \tag{35}$$

in the same way, when Eq. (27) is multiplied by $\frac{\eta_{\perp,S}}{N_S(z)}$ and $\frac{\eta_{II,P}}{N_P(z)}$, we can derive Eqs. (36) and (37) respectively.

$$X_S(z_j, z_k, t) = \frac{R_P^{-1}(z_j,t)-R_P^{-1}(z_k,t)}{R_\delta(z_j,t)-R_\delta(z_k,t)}, \tag{36}$$




$$X_P(z_j, z_k, t) = \frac{R_S^{-1}(z_j,t) - R_S^{-1}(z_k,t)}{R_\delta^{-1}(z_j,t) - R_\delta^{-1}(z_k,t)} \tag{37}$$

with time $t$. Here it can be noted that the influence of the cross-talk factor $\varepsilon_r$ is also removed. By averaging many measurements of $X_\delta(z_j, z_k, t)$, we obtained a mean value for $X_P$ (Eq. (30)) or $X_S/X_P$. Similarly, evaluation of many values of $X_S(z_j, z_k, t)$ yields an accurate estimate for $X_S$ (Eq. (29)) or $X_P X_\delta$ (when combining Eqs. (28) and (30)), and the same holds for the analysis

of many $X_P(z_j, z_k, t)$ values to obtain a trustworthy observation of $X_P$ (Eq. (28)) or $X_S/X_\delta$ (see Eq. (29) and (30)). With other words, the constant $X_\delta$ can be determined directly from Eq. (35), or calculated from the constants $X_P$ and $X_S$ obtained with Eqs. (36) and (37). As will be shown below (and in the result section), the constants $X_P$, $X_S$, and $X_\delta$ are used to simultaneously determine the volume depolarization ratio in three different ways.

Given the form of Eqs. (35)-(37), observable differences between the height points $z_j$ and $z_k$ are needed for its evaluation, in

practice, only altitude regions should be selected in the determination of $X_P$, $X_S$, and $X_\delta$ where significant changes in the depolarization ratio occur, e.g., in liquid-water clouds where multiple scattering by droplets produce steadily increasing depolarization with increasing penetration of laser light into the cloud (Donovan et al., 2015; Jimenez et al., 2017; Jimenez et al., 2018).

To derive now the linear depolarization ratio, we divide Eq. (24) by Eq. (25),

$$\frac{N_S}{N_P} \frac{\eta_{II,P}}{\eta_{\perp,S}} = X_\delta R_\delta = \frac{(1+\varepsilon_r)(1+\varepsilon_l) - \frac{(1-\delta)}{(1+\delta)}(1-\varepsilon_l)(1-\varepsilon_r)\cos(2\theta)}{(1+\varepsilon_r)(1+\varepsilon_l) + \frac{(1-\delta)}{(1+\delta)}(1-\varepsilon_l)(1-\varepsilon_r)\cos(2\theta)}. \tag{38}$$

Furthermore, we introduce the total cross-talk factor $\xi_{tot}$,

$$\xi_{tot} = \frac{(1+\varepsilon_r)(1+\varepsilon_l)}{(1-\varepsilon_l)(1-\varepsilon_r)\cos(2\theta)} \geq 1, \tag{39}$$

which takes account for the combined effect of the emitted elliptically polarized wave front $\varepsilon_l$, of the angular misalignment between emitter and receiver (described by the rotation angle $\theta$), and of the cross-talk between receiver channels described

by $\varepsilon_r$. The factor $\xi_{tot}$ would be equal to 1 if the emitted laser pulses are totally linearly polarized, misalignment of the receiver unit could be avoided, and cross talk between receiver channels are negligible.

Now Eq. (38) can be rewritten after dividing the numerator and denominator by $(1-\varepsilon_l)(1-\varepsilon_r)\cos(2\theta)$ and rearranging the equation:

$$X_\delta R_\delta = \frac{\xi_{tot} - \frac{(1-\delta)}{(1+\delta)}}{\xi_{tot} + \frac{(1-\delta)}{(1+\delta)}} \tag{40}$$

and the volume depolarization ratio can be obtained from Eq. (40) after rearrangement,

$$\delta(R_\delta, X_\delta, \xi_{tot}) = \frac{1 - \xi_{tot} + X_\delta R_\delta (1+\xi_{tot})}{1 + \xi_{tot} + X_\delta R_\delta (1-\xi_{tot})}. \tag{41}$$

As shown in Eq. (41), the volume depolarization ratio can be calculated by using the ratio $R_\delta$ between the cross and co-polarized signals and when the constants $X_\delta$ and $\xi_{tot}$ are known. In the first step, the inter-channel constant $X_\delta$ has to be measured. Then $\xi_{tot}$ can be estimated in a region (defined by height $z_{mol}$) with dominating Rayleigh backscattering for which





the volume depolarization ratio, $\delta_{mol}$, is known. Behrendt et al. (2002) estimated theoretically a value of the linear depolarization ratio caused by molecules of 0.0046 for a lidar system whose interference filters have a FWHM=1.0 nm, however, Freudenthaler et al. (2016b) has found a value of $0.005 \pm 0.012$ based on long-term measurements in aerosol and cloud-free tropospheric height regions, we used this value and we have considered the propagation of this systematic uncertainty in our calculations. From, Eq. (41) $\xi_{tot}$ is given by.

$$\xi_{tot} = \left(\frac{1-\delta_{mol}}{1+\delta_{mol}}\right)\left(\frac{1+CR_\delta(z_{mol})}{1-CR_\delta(z_{mol})}\right), \tag{42}$$

By calculating the ratio between Eqs. (24) and (26) (co to total) or the ratio between Eqs. (25) and (26) (cross to total), the volume depolarization ratio can also be derived:

$$\delta(R_S, X_S, \xi_{tot}) = \frac{1-\xi_{tot}(1-2X_S R_S)}{1+\xi_{tot}(1-2X_S R_S)}, \tag{43}$$

$$\delta(R_P, X_P, \xi_{tot}) = \frac{1-\xi_{tot}(2X_P R_P-1)}{1+\xi_{tot}(2X_P R_S-1)}. \tag{44}$$

In summary, the volume linear depolarization ratio can be calculated after the determination of the constants $X_P$, $X_S$, $X_\delta$ and $\xi_{tot}$. Then the signal ratio profiles $R_P(z)$, $R_S(z)$, and $R_\delta(z)$ are required and calculated within Eqs. (31), (32) and (33), and by considering Eqs. (41), (43) and (44) the depolarization ratio can be finally calculated by using either the pair of signals $N_S$ and $N_P$, the pair $N_S$ and $N_{tot}$, or the pair $N_P$ and $N_{tot}$ respectively. However, the expected errors in the retrievals are not the same for all this pairs, since they present different sensitivities to changes in the depolarization ratio, obtaining the largest uncertainties when the pair $N_P$ and $N_{tot}$ is used.

## 4 Observations

### 4.1 Application of the calibration approach to a measurement case

To test the method introduced in Sect. 3, the measurement case from 19 September 2017 was analyzed and the results are presented in this section. Figure 4 provides and overview of the atmospheric situation. An aerosol layer reached up to about 2.8 km height and was topped by a persistent, shallow altocumulus deck with cloud base height at 2.6-2.7 km a.g.l. (about ground level)

Although the time resolution of the lidar measurements is 30 seconds, to reduce computing time and signal noise, we consider 5 minutes average measurements. Figure 5 shows as example the three range-corrected signals of the polarization lidar, the signal ratios as defined by Eqs. (31)-(33), and the corresponding inverse ratios for a 5-minute measurement.




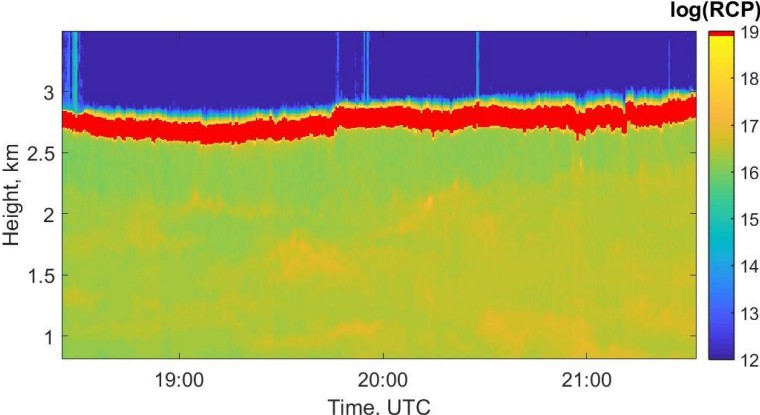

**Figure 4: Range-corrected 532 nm total backscatter signal (RCP) measured on 19 September 2017 with 30 s and 7.5 m vertical resolution.**

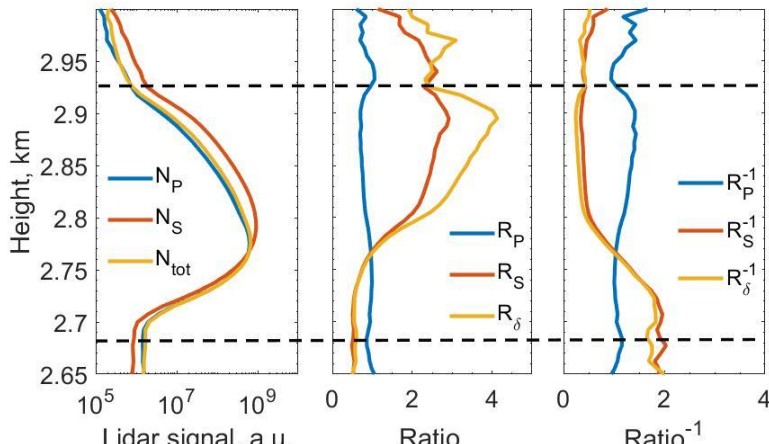

**Figure 5: Example of a 5-minute profile of range-corrected lidar signals from the channels, signal ratios, and inverse ratios. The calibration procedure considers all signals of the 3-hour measurement period shown in Fig. 4. The dash line indicates the range where the calibration calculations where done.**

10    In the next step of the data analysis and calibration procedure, we selected the height range from a few meters below cloud base up to 240 meters above cloud base for each 5-minute averaging period $t$, then we computed the instrumental inter-channel ratios $X_P(z_j, z_k, t)$, $X_S(z_j, z_k, t)$, and $X_\delta(z_j, z_k, t)$ with Eqs. (37), (36), and (35), respectively. Height resolution was 7.5 m. The result is shown in Fig. 6.





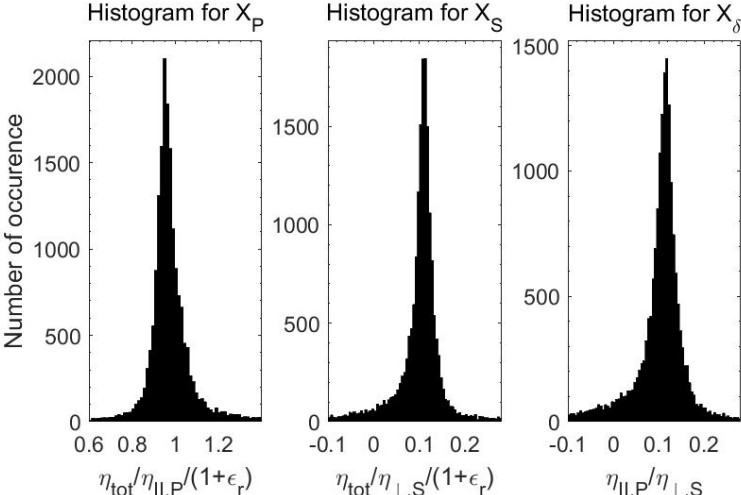

**Figure 6: Histograms for the inter-channel constants $X_P$, $X_S$ and $X_\delta$. Each point corresponds to a combination $z_j$ and $z_k$ in a 5-minute period, obtaining about 16000 data points for this measurement case.**

The mean values of the constants with the respective statistical error based on Fig. 6 are: $X_P = 0.965 \pm 0.012$, $X_S = 0.108 \pm 0.005$ and $X_\delta = 0.110 \pm 0.006$. The reason for these low uncertainties is that the calibration is performed in a cloudy region so that every channel shows high count rates and thus high signal-to-noise ratios.

By using constant $X_\delta$ and Eq. (42), a mean value of $\xi_{tot} = 1.118 \pm 0.008$ was obtained. The cross-talk factor has a large impact on the retrieval of the volume linear depolarization ratio only in the region with low depolarization ratios. Table 1 summarize these instrumental constants.

**Table 1: Values of the instrumental inter-channel constants and cross-talk factor determined for the measurement case presented.**

| Instrumental constant | Value |
|:---:|:---:|
| $X_P$ | $0.965 \pm 0.012$ |
| $X_S$ | $0.108 \pm 0.005$ |
| $X_\delta$ | $0.110 \pm 0.006$ |
| $\varepsilon_{tot}$ | $1.118 \pm 0.008$ |

Figure 7 presents the computed height profiles of the volume linear polarization ratio computed by means of Eqs. (41), (43), and (44). Good agreement between the different solutions is visible. However, the depolarization ratios obtained from the channels $N_P$ and $N_{tot}$ (blue) shows the largest uncertainties. The profile-mean absolute uncertainties in $\delta(R_{II}, X_P)$, $\delta(R_\perp, X_S)$ and $\delta(R_\delta, C)$ are 0.034, 0.0139 and 0.0137, respectively. The three derived depolarization ratios agree well in the cloud region, differences appear in the upper cloud part caused by strongly reduced count rates.





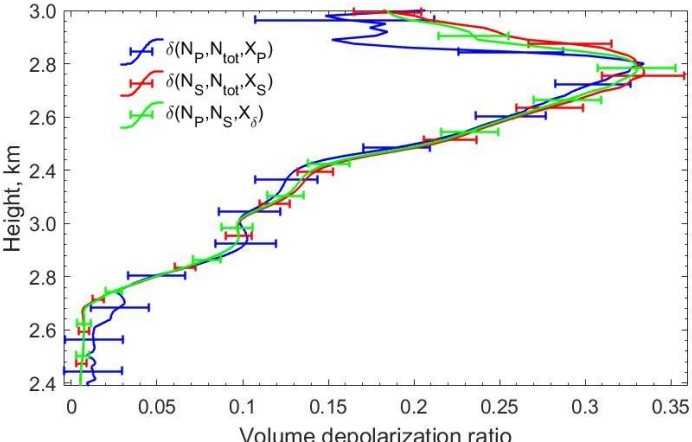

**Figure 7: Profiles of the volume linear depolarization ratio for the 3 hours period in the cloud region, using the three pairs of signal ratios presented in Eqs. (41), (43) and (44). The error bars include the statistical and systematical uncertainties.**

5    Figure 8 presents the volume depolarization ratio with 30 s temporal resolution. The signal ratio $R_\delta$ and the constant $X_\delta$ where used. These profiles are the basis for the retrieval of the microphysical properties of the liquid-water cloud. The results will be dicussed in a follow-up article (Jimenez et al., 2018b).

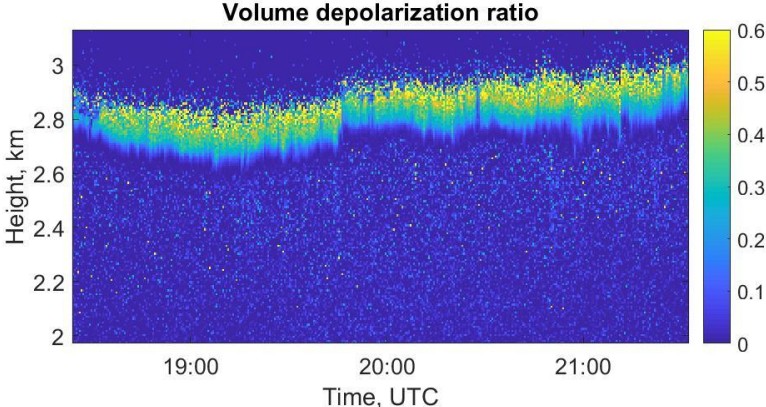

**Figure 8: Volume linear depolarization ratio for the entire 3-hour period, shown in Fig. 4. The temporal resolution is 30 seconds.**

Figure 9 shows a comparison between the measurements of the volume linear depolarization ratio with the lidar systems MARTHA and BERTHA (Backscatter Extinction Lidar Ratio Temperature and Humidity profiling Apparatus) (Haarig et al., 2017) which was located about 80m from the MARTHA system. The observations were conducted at Leipzig (51°N, 12°E) during an event with a dense biomass burning smoke layer in the stratosphere on 22 August 2017 (Haarig et al., 2018).  Very

15    good agreement was obtained.





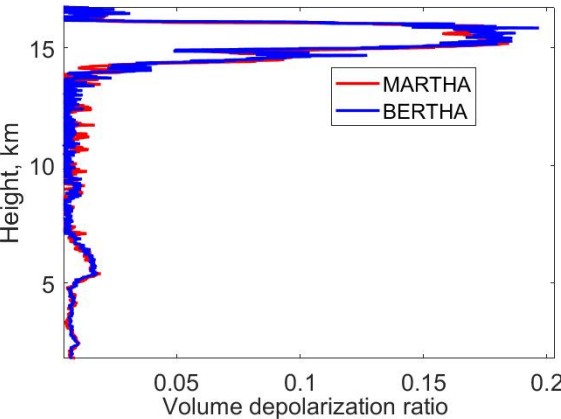

**Figure 9: Volume linear depolarization ratio obtained with MARTHA (3-signal method) and BERTHA ($\pm45°$ method) on 22 August 2017 (Haarig et al., 2018). The systems were calibrated independently.**

## 4.2 Long-term stability of the polarization lidar calibration and performance

The time series of the inter-channel constant $X_\delta$ obtained from MARTHA observations between day 120 and 320 of 2017 is presented in Fig. 10. The respective time series of $\xi_{tot}$ is given in Fig. 11. As can be seen, the calibrations values show the lowest uncertainties in the inter-channel constants (of about 4%) when altocumulus layers with a stable cloud base and moderate light extinction were present. Higher uncertainty levels were observed in the case of cirrus clouds (green, 11%) and the Saharan dust layer. In the case of very thick cumulus clouds (black), the mean uncertainty was 21%. One reason for these differences in the uncertainty of $X_\delta$ is that the system was optimized for the observation of low-altitude liquid-water clouds. The selected large attenuation of the channels prohibited an optimum detection of high-level dust layers and ice clouds. Furthermore, liquid clouds are favorable for calibration because the volume depolarization ratio increases very smoothly as a result of the increasing multiple scattering impact. At these conditions, a large number of measurement pairs for heights $z_j$ and $z_k$ with different depolarization ratios are available. Some slight changes of $X_\delta$ occurred when the attenuation configuration of the polarization receivers was changed. Small day-to-day changes were caused by small variations in the response of each detector with time.

In Fig. 10 are the retrieved values of $\xi_{tot}$, small variations can be seen but they remain much lower than the uncertainties, and no stronger variations can be noted with changes in the attenuation or changes of the calibration medium (water cloud, cirrus, Saharan dust layer). In 2017, the mean value $\xi_{tot} = 1.109 \pm 0.009$.





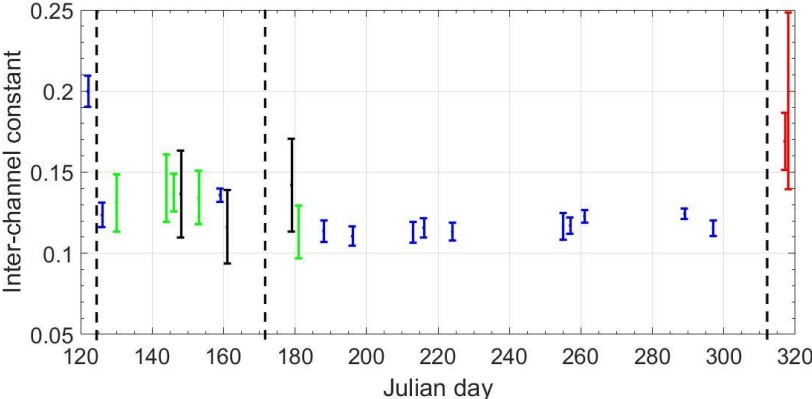

**Figure 10: Time series of the inter-channel calibration constant $X_\delta$ measured from end of April to mid November 2017. The vertical bars show the uncertainty in the retrieval. The calibration procedure was based on lidar measurements in liquid-water clouds (blue), cirrus clouds (green), during optically thick cumulus events (black), and Saharan dust periods (red). The dash lines indicate the days where changes in the attenuation configuration of the channels were made.**

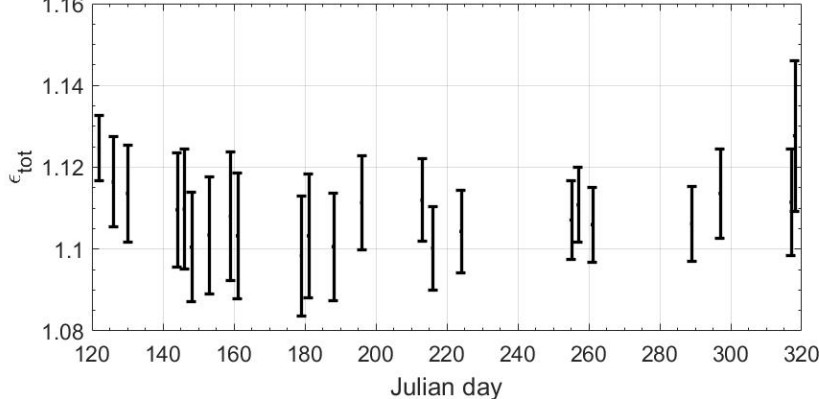

**Figure 11: Time series of the total cross talk factor $\varepsilon_{tot}$ measured in 2017. The vertical bars show the uncertainty in the retrieval, which include the statistical error from the determination of the inter-channel constants and systematical errors from the value considered in the molecular region $0.005 \pm 0.0012$.**





## 5 Summary and conclusions

In this work a new formalism to calibrate polarization lidar systems based on three detection channels has been presented. We propose a simple lidar polarization receiver, based on three telescopes with a polarization filter on the front (in the case of the cross and co polarized channels), this set up removes the effect of the receiver optics on the polarization state of the collected

backscattered light, simplifying the measurement concept. The derivation of the volume linear depolarization ratio considering the instrumental effects on the proposed system was described in section 3, here there are three effects considered: the emitted laser beam (after beam expander) is slightly elliptically polarized ($\mathcal{E}_l$), there is an angular misalignment ($\theta$) of the receiver unit with respect to the main polarization plane of the emitted laser pulses and there is a small cross-talk amount in the detection channels (co and cross) ($\mathcal{E}_r$). These instrumental parameters can be summarized into one single constant, the so-called total

cross-talk ($\xi_{tot}$).

The methodology permits the determination of the so-called inter-channel constants $X_P$, $X_S$ and $X_\delta$, which depend on the attenuation and detector response of each channel, and thus it is expected to vary between different measurement days. The calibration is based on actual lidar measurement periods, providing large amount of input data for accurate estimation of the mean value of the instrumental constants. However, it needs a strong depolarizing medium for its application, e.g., water

clouds.

A case study of a liquid-water cloud observation was presented, the 3-hours period demonstrates the potential of the new technique for the retrieval of accurate high temporal resolution depolarization profiles. The method is simple to implement and allows high quality depolarization ratio studies. Long term studies indicated the robustness and stability of the three-signal lidar system over long time periods.  A comparison with another polarization lidar shows excellent agreement regarding the

derived volume linear polarization ratio of biomass burning smoke throughout the troposphere and the lower stratosphere up to 16 km height.

Generally, the volume of the depolarization ratio does not depend on the field of view of the receiver, however in multiple scattering regime (e.g. in liquid water clouds), it does, and it depends additionally on the microphysical properties of the cloud. In the next articles we will focus on the use of this 3-signal approach in our new technique to retrieve microphysical properties

of liquid water clouds from depolarization measurements at two receiver field of views.



**Appendix A: General case regarding the rotation of the polarization filters with respect to the true polarization axis of the emitted light**

For the derivation outlined in Section 3 it is assumed that the polarization filters in front of the cross- and co-polarized telescopes are pointing 90° with respect to each other. However, in the general case, when their angular deviation with respect to their respective components is different ($E_P$ to $E_{II}$ and $E_S$ to $E_\perp$), Eqs. (24) and (25) have a different angular component. In this appendix we analyze this general case and discuss the need of implementation depending on the results obtained.

We define the angles $\theta_P$ and $\theta_S$ as the angular misalignment of the channels $E_P$ and $E_S$ with respect to $E_{II}$ and $E_\perp$ respectively

(see Fig A1). We rewrite Eqs. (24) - (26), we factorize by $(1 + \varepsilon_r)$ and to simplify the expression, we adopt the polarization parameter $a = \frac{(1-\delta)}{(1+\delta)}$ again.

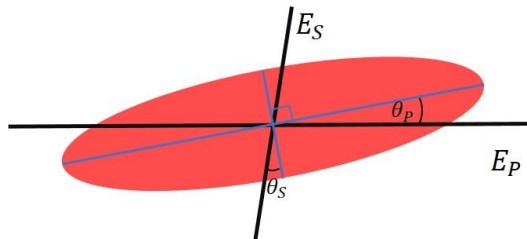

**Figure A1: Scheme of the observation of the polarization state of the backscattered light (similar to Fig. 3), the co and cross channels are misaligned with respect to their components in an angle $\theta_P$ and $\theta_S$ respectively.**

$$\frac{N_P(z)z^2}{\eta_{II,P}P_0T^2(z)} = \beta_{II,in}(z) + \varepsilon_r\beta_{\perp,in}(z) = (1 + \varepsilon_r)\left(1 + a(z)\frac{(1-\varepsilon_l)}{(1+\varepsilon_l)}\frac{(1-\varepsilon_r)}{(1+\varepsilon_r)}\cos(2\theta_P)\right)\beta(z)/2, \tag{A1}$$

$$\frac{N_\perp(z)z^2}{\eta_{\perp,S}P_0T^2(z)} = \beta_{\perp,in}(z) + \varepsilon_r\beta_{II,in}(z) = (1 + \varepsilon_r)\left(1 - a(z)\frac{(1-\varepsilon_l)}{(1+\varepsilon_l)}\frac{(1-\varepsilon_r)}{(1+\varepsilon_r)}\cos(2\theta_S)\right)\beta(z)/2, \tag{A2}$$

$$\frac{N_{tot}(z)z^2}{\eta_{tot}P_0T^2(z)} = \beta_{in}(z) = \beta(z). \tag{A3}$$

In a similar way as we defined $\xi_{tot}$, we define the total cross-talk factor for the co and cross polarized channels respectively.

$$\xi_P = \frac{(1+\varepsilon_r)(1+\varepsilon_l)}{(1-\varepsilon_l)(1-\varepsilon_r)\cos(2\theta_P)} \geq 1, \tag{A4}$$

$$\xi_S = \frac{(1+\varepsilon_r)(1+\varepsilon_l)}{(1-\varepsilon_l)(1-\varepsilon_r)\cos(2\theta_S)} \geq 1, \tag{A5}$$

The three-signal polarization equation (Eq. (27)) can be rewritten in a general form, when adding Eqs. (A1) and (A2) and considering Eq. (A3):





$$\left(\frac{N_P(z)}{\eta_{II,P}} + \frac{N_S(z)}{\eta_{\perp,S}}\right) = (1 + \varepsilon_r)(1 + a(z)(\xi_P^{-1} - \xi_S^{-1})/2)\frac{N_{tot}(z)}{\eta_{tot}}. \tag{A6}$$

The term $\xi_P^{-1} - \xi_S^{-1}$ depends on the difference of the cosines of $2\theta_P$ and $2\theta_S$, we define the parameter $\xi_S^P$ which account for the difference of the impact of the polarization channels.

$$\xi_S^P := (\xi_P^{-1} - \xi_S^{-1})/2 = \frac{(1-\varepsilon_l)(1-\varepsilon_r)}{2(1+\varepsilon_l)(1+\varepsilon_r)}(\cos(2\theta_P) - \cos(2\theta_S)). \tag{A7}$$

This factor can be positive or negative, depending on which polarization filter is more misaligned, and it is equal to zero when they point 90° with respect to each other. Eq. (A6) can be expressed as:

$$\left(\frac{N_P(z)}{\eta_{II,P}} + \frac{N_S(z)}{\eta_{\perp,S}}\right) = (1 + \varepsilon_r)(1 + \xi_S^P a(z))\frac{N_{tot}(z)}{\eta_{tot}}, \tag{A8}$$

We adopt the notation:

$$\Delta R_P(z_j, z_k) = R_P(z_j) - R_P(z_k), \tag{A9}$$

for account the difference between the signal ratios $R_P$, $R_S$ and $R_\delta$, between the polarization parameter $a$, and between the ratios $a/R_S$ and $a/R_P$ at the heights $z_j$ and $z_k$. In an equivalent way as we derived Eqs. (35)-(37) we can obtain a general solution for the instrumental inter-channel constants:

$$X_P(z_j, z_k) = \frac{\Delta R_S^{-1}(z_k) + \xi_S^P \Delta\left(\frac{a}{R_S}(z_j, z_k)\right)}{\Delta R_\delta^{-1}(z_j, z_k)}, \tag{A10}$$

$$X_S(z_j, z_k) = \frac{\Delta R_P^{-1}(z_j, z_k) + \xi_S^P \Delta\left(\frac{a}{R_P}(z_j, z_k)\right)}{\Delta R_\delta(z_j, z_k)}, \tag{A11}$$

$$X_\delta(z_j, z_k) = \frac{\Delta R_P(z_j, z_k) + \xi_S^P \frac{\Delta a(z_j z_k)}{2X_P}}{\Delta R_S^{-1}(z_j, z_k)}, \tag{A12}$$

In an absolute sense it would not be possible to determine the inter-channel constants $X_P$, $X_S$ and $X_\delta$ without knowing the polarization parameter (or the depolarization ratio), however, the impact on Eqs. (A10) - (A12) of their respective second term can be very small, since it depends on the difference of the cosines of small angles. For example, if $2\theta_P = 5°$ and $2\theta_S = 10°$, using Eq. (A7) $\xi_S^P = 0.005\frac{(1-\varepsilon_l)(1-\varepsilon_r)}{(1+\varepsilon_l)(1+\varepsilon_r)}$. Considering this small effect, a first guess of the polarization parameter would be

sufficient to solve equations (A10) - (A12).

Calculating the three ratios between Eqs. (A1), (A2) and (A3), we can obtain the volume linear depolarization ratio, similarly as how it was done for Eqs. (41), (43) and (44).

$$\delta(R_S, X_S, \xi_P) = \frac{1 - \xi_P(1 - 2X_S R_S)}{1 + \xi_P(1 - 2X_S R_S)}, \tag{A13}$$

$$\delta(R_P, X_P, \xi_S) = \frac{1 - \xi_S(2X_P R_P - 1)}{1 + \xi_S(2X_P R_P - 1)}, \tag{A14}$$

$$\delta(R_\delta, X_\delta, \xi_P, \xi_S) = \frac{1 + \frac{\xi_S}{\xi_P}X_\delta R_\delta - \xi_S(1 - X_\delta R_\delta)}{1 + \frac{\xi_S}{\xi_P}X_\delta R_\delta + \xi_S(1 - X_\delta R_\delta)}. \tag{A15}$$




$$a = \frac{1-\delta}{1+\delta},$$ (A16)

In the measurement example presented, we performed an iterative computation procedure to determine the inter-channel calibration constants and the cross-talk factors. Using Eqs. (A10)-(A12), in a first run we determined the inter-channel constants when we assume $\xi_S^P = 0$, i.e. $\theta_P = \theta_S$, a first guess of the volume depolarization ratio using each pair of signals is

obtained (Eqs. (A13)- (A15)), then the corresponding cross-talks $\xi_P$ and $\xi_S$ are determined by imposing a mean value of $\delta = 0.005 \pm 0.012$ in the free aerosol region (Freudenthaler et al., 2016b). The second run takes the values of $\xi_S^P \neq 0$ and of the polarization parameter $a(z,t)$ (Eqs. (A14) and (A16)) from the first run and the inter-channel constants are computed again. Figure A2 shows the results of performing the calibration iteratively. Small differences between the values obtained in the first and second run can be noted, in fact, the variations are smaller than the error of the respective constants, and we can see that

after the second run, all values remain practically constant. The mean values of the instrumental constants after 6 iterations are listed in the Table A1.

In this measurement case we found a value for $\xi_S^P = -0.008$. Due to this small value there are no important variations between the first guess and the second run, therefore we conclude that by assuming $\xi_S^P = 0$ a fast and practical inversion procedure is possible. However, in cases with larger differences between $\theta_P$ and $\theta_S$, an iterative procedure as described above would be

needed.

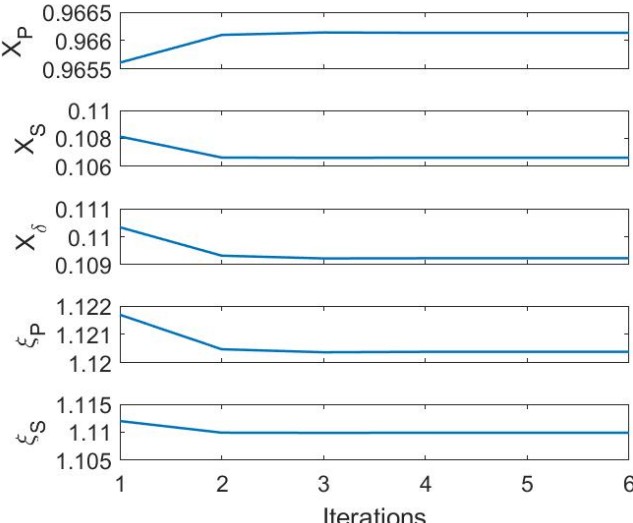

**Figure A2: Instrumental channels obtained with an iterative procedure. We did not include the error bars since they are much larger than the variations between runs.**



**Table A1: Results of the instrumental constants after using the iterative procedure (6 runs)**

| Instrumental constant | Value |
|:---:|:---:|
| $X_P$ | $0.966 \pm 0.011$ |
| $X_S$ | $0.106 \pm 0.005$ |
| $X_\delta$ | $0.109 \pm 0.006$ |
| $\xi_P$ | $1.120 \pm 0.007$ |
| $\xi_S$ | $1.110 \pm 0.007$ |

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
