# Peer review of "Polarization lidar: An extended three-signal calibration approach"

_Atmospheric Measurement Techniques, 2018_

## Referee Comment (RC1) · Anonymous Referee #1 · 24 Dec 2018

**General Comments:**

The authors are presenting the mathematical formulation behind an extended approach of the three-signal technique, developed for calibrating depolarization lidar instruments. Complementary to the theoretical approach, the authors are presenting also some experimental results, verifying in that way the stability and efficiency of their method. The manuscript is well written, acknowledging previous relevant studies, and has strong scientific merit. Therefore, in my opinion worth of being published in the Atmospheric Measurement Techniques journal. In order to be improved I would kindly suggest to the authors to take into consideration the following specific comments.

**Specific Comments:**

1. Page 3, Line 9: "…consists of 2''achromatic lens…". Consider providing in a parenthesis also the equivalent of the 2'' in units of mm.
2. Page 3, Line 13: Is the value of 650 m theoretically calculated or experimentally measured? In any case consider providing a reference at this point.
3. Page 5, Line 26: Maybe "described" is more appropriate than "considered".
4. Figure 3: Consider annotating this figure with the letters (a), (b), (c), (d), in accordance to the figure legend. Moreover, make clear also in the figure that (a) and (b) refer to the emission while (c) and (d) to the receiving units.
5. Page 7, Line 13: "…for the calibration is to insert an additional polarization…"
6. Page 7, Line 21: "…, Müller matrices representing…"
7. Page 7, Line 27: "$P_o$ is the number of emitted laser photons…"
8. Page 8, Line 6: Consider replacing the sub scripts with capital characters, in order to be consistent with the annotation followed in the manuscript.
9. Page 9, Line 12: "…, but depends on the receiver…"
10. Page 10, Line 14: Eq. 38 is the product of Eq. 25 divided by Eq. 24, and not the inverse.
11. In Eq. 42 I am missing the information of the variable C. Please specify to which quantity C refers to.
12. Page 11, Line 20: "… provides an overview of …"
13. Page 11, Line 21: "(above ground level)"
14. Consider presenting clearer the x-label of Figure 6 (e.g. $\eta_{tot} / \left[ \eta_{\parallel,P} \cdot (1 + \varepsilon_r) \right]$)
15. I would kindly suggest to the authors to show the profiles presented in Figure 7 up to higher altitudes (e.g. 4 km). Moreover, the profiles obtained by the ratio $N_P / N_{tot}$, compared to the rest two, seems to demonstrate greater variability with atmospheric height, in a way that I would say artificial layers are introduced. This can be seen for atmospheric heights inside the water cloud but also below (2.4 - 2.7 km). Is this also a result of the low SNR, even though

that the profiles refer to 3 hours of measurement period? In any case the authors are kindly requested to comment on this.

16. Figure 9 is very important. Legend: My suggestion is to use the phrase "(extended 3-signal method)". Additionally, it would be beneficiary for the manuscript if in the same figure, the profile of volume depolarization obtained by the conventional 3-signal technique (Reichard et al., 2003), is also shown. This will clearly demonstrate the improvement achieved by following the extended method proposed here, which takes into account various types of instrumental effects (e.g. the not perfectly polarized emitted laser light).

17. The manuscript contains many equations and variables and this may easily confuse a reader. Therefore, I would kindly suggest to the authors to list all the variables used in a table (Appendix section), along with a small phrase describing them.

---

## Referee Comment (RC2) · Anonymous Referee #3 · 26 Dec 2018

**General Comments**

The manuscript describes a formalism to calibrate the polarization channels of a lidar instrument by considering several sources of systematic uncertainties. This is a really important work since not many publications are dealing with the three signal calibration approach. Even if the method is commonly used by several lidar stations, a complete theoretical support with experimental results was somehow lacking. Under these considerations I consider the manuscript of scientific relevance and suggest to be published in AMT after some revisions. In order to be improved I would kindly suggest the authors to consider several comments stated in the following section:

**Specific comments:**

1. Abstract: "A comparison with another polarization lidar" replaced by "A comparison with a second polarization lidar"

2. Page 2, Line22-25: Please use the same notation for the sections. Currently we can find "section", "Section", "Sect."

3. Section 3.1 starts with the statement that the manuscript follows the "notation and explanations of Freudenthaler (2016)" from AMT. Still in the following description, the authors define the misalignment between the polarization axis of the transmitted light and the co-polarized receiver channel as $\theta$ - Page 5, Line 25-26: "The misalignment between the polarization axis of the transmitted light and the co-polarized receiver channel (defined by the respective polarization filter in front of the PMT) is characterized by angle $\theta$ ...."

   The corresponding parameter in Freudenthaler (2016) should be the "Rotation of the plane of horizontal linear polarisation of the laser around the z axis (laser rotation)" which is relative to the receiving unit reference plane. Since the manuscript refers to a similar study performed by a lidar station from the same research network EARLINET-ACTRIS, I would suggest the authors to use the same notation used in the previous work ($\alpha$). Keeping the same variable names and notations as used in previous studies will help a reader familiar with similar studies and encourage the use of standardized variables and parameters.

4. Same comment as above applies for Page 6, Line 17: "The rotated polarization axis is represented in Fig. 3c, and after" and Figure 3 (also Page 8, Line 18).

5. Page 7, Line 13: "A commonly used method for the calibration is  to insert"

6. Page 7, Line 27: "$P0$ is the  number of emitted laser photons"

7. Page 9, Line 25: This section should be described in more detail and the reasoning behind the use of two altitude heights should be clearly mentioned. Please consider extending this section since it is an important part of the theoretical background required to use the calibration technique used in this study.

8. Page 10, Line 3: "we obtained a mean value for $X_\mathrm{p}$". Is this really "p" or is this "$\delta$"?

9. Page 10, Line 3-9: "Similarly, evaluation of many values of ..... are used to simultaneously determine the volume depolarization ratio in three different ways.". This section should be described in much more detail. Even if most of the readers are experts in lidar techniques, they are not familiar with the theoretical description and formalism presented in the manuscript. The theory behind this calibration technique is really valuable since this is one of the few manuscripts dealing with the three channel calibration topic and it is important to provide a complete set of information on the theory. This section must be reconsidered before the manuscript is send for publication.

10. Page 10, Line 14: "To derive  the linear depolarization"

11. Page 10, Line 28: "In the first step, the inter-channel constant $X_\delta$ has to be measured." More detail must be provided by the authors. The experimental technique on how to perform the assessment of $X_\delta$ must be provided since this is one of the key parameters for the calibration of the depolarization channels.

12. Eq. 42: please give more details on the missing variable "C".

13. Page 13, Line 8: "By using constant and Eq. (42), a mean value of ..." Please provide more information on this topic.

14. Page 14, Line 10-15: Since the comparison between the volume linear depolarization ratio measured by MARTHA and BERTHA is designed to validate the calibration technique used in this study, I would advise to also use a second case for this comparison. A strong depolarizing layer (e.g mineral) would help validate the results for highly depolarizing layers.

15. Page 17, Line 22-25: I do not see the necessity of this section. A link with further studies was already included in the introduction of the study and since this section is not connected to the conclusions I would advice to remove it for the final version of the manuscript.

16. Since the manuscript has an important theoretical section containing many variables and equations, I would suggest adding an additional list of variables containing a comprehensive description for each element. Please consider following the same terminology used by Freudenthaler (2016)

---

## Referee Comment (RC3) · Anonymous Referee #2 · 26 Dec 2018

**General comments**

In this manuscript (amt-2018-370), authors describes the three-signal depolarization calibration for lidar systems proposed by Reichardt et al., 2003 by means of the Stokes-Müller formalism. This allowed them to take into account some lidar polarizing effects. Additionally, authors presented the experimental setup and some results (a case study and an analysis of the temporal evolution of the depolarization calibration) to validate the procedure. Both the theoretical and experimental basis are well described in a well-written and structured paper. Since lidar depolarization seems to be in the spotlight of atmospheric science due to its application in aerosol/cloud microphysical retrievals (and thus, in aerosol-cloud interaction), I would recommend its publication whether the following comments are considered by the authors.

A small discussion about the advantages and disadvantages of this method, in comparison with others already implemented, would be extremely useful for the community as far as lidar design is concerned.

**Specific comments**

Page 4, line 3: *'In the alignment process, the cross-polarized axis is found when the count rates are at the minimum.'*

This process seems to be inaccurate because the change of the signal due to several degrees can be masked by the signal noise. Did you check the accuracy of this procedure? Could you provide the uncertainty?

Page 4, line 18: *'Based on this theoretical framework we will derive three lidar equations for our three measured signal components.'*

I think that the theoretical framework has not been presented yet. Please, considered to change by 'Based on the theoretical framework of ___, we will derive […]'.

Page 5, lines 10-14: *'In our approach, …'*

I recommend to mention the Figure 3 somehow. It will be easier to understand this paragraph following the steps of Figure 3.

Page 5, line 18: *'We introduce the so-called cross-talk term $\varepsilon_i$'*

In the line 1 of the same page, it is stated that the notation and explanations of previous manuscripts are used. However, the term 'cross-talk' is mainly used to describe non-ideal beam-splitter cubes instead of the depolarization of the outgoing laser light (emitting block). If I properly understood, the cross-talk term would correspond to the depolarisation of the laser light after crossing the transmission block (a king of linear polarisation parameter $a_L$, according to the Freudenthaler's paper). Additionally, I would say that the angle $\theta$ in this manuscript corresponds to the angle $\alpha$ in the Freudenthaler's paper. For the sake of clarity, it would be very

helpful for the community whether the same nomenclature is used or, at least, a small mention about the connections is included.

Page 6, line 2 and Eq (6):

According to the Freudenthaler's paper, the rotation $R(\theta)$ proposed in this manuscript is a particular case of the emitting block, being other polarizing effects omitted such as the diattenuation. It would be helpful for the readers to have a list of parameters considered ideals.

Page 6, Figure 3:

It would be helpful for future readers to link each step in the figure with the term, as follow:

[Figure]

Page 7, line 13: *'A commonly used method for the calibration is the to insert an extra polarization filter[…]'*

Typo?

Page 8, line 6: *'Because identical polarization filters are used in our lidar setup, we can assume $Dp = Ds^{-1}$.'*

What do you mean with 'identical'? Even the same model of polarizer made by the same company might show quite different behaviors. Additional details should be addressed to support this sentence.

Page 9, line 2: *' […] based on a measurement example, we demonstrated that the impact of this assumption can be neglected in our system.'*

Could you provide any indication about the validity of this assumption in other systems?

Page 10, line 11: *'liquid-water clouds where multiple scattering by droplets produce steadily increasing depolarization with increasing penetration of laser light into the cloud […]'* and Page 17, line 21: *'the volume of the depolarization ratio does not depend on the field of view of the receiver, however in multiple scattering regime (e.g. in liquid water clouds), it does […]'.*

Could the multiple scattering be a problem for the depolarization calibration?

Page 10, line 29: '*Then $\xi tot$ can be estimated in a region (defined by height $zmol$) with dominating Rayleigh backscattering […]*'

This the most important handicap I detect in this method. $\xi tot$ must be estimated in a particle-free region where the SNR used to be quite low. This is the same handicap of the classical molecular calibration, including a more complicate ldiar system since three channels are required instead of two. So, why is this method more advisable?

Page 11, Eq. (42):

Which is the meaning of the term 'C'? I was not able to find its definition. From Eq. (41) to Eq. (42), I got that $C = X_\delta$. Please, specify it.

Page 12, line 10 and caption of Figure 6:

Whereas it is stated that the height range goes from a few meters below the cloud base up to 240 m above (page 12 line 10) in the caption of the Figure 6, it is noted that 16000 data points were obtained. Could you explain the huge number of data points in this small height range?

Page 13, line 8: '*The cross-talk factor has a large impact on the retrieval of the volume linear depolarization ratio only in the region with low depolarization ratios.*'

Please, include whatever is necessary to demonstrate this sentence.

Page 15, line 8: '*4.2 Long-term stability of the polarization lidar calibration and performance*' and Page 17, line 18: '*Long term studies indicated the robustness and stability of the three-signal lidar system over long time periods.*'

The calibration stability was analyzed between April and November 2017 (8 months). I would use 'long-term' for larger periods and thus, I suggest replacing 'long-term stability' by 'temporal stability'.

Page 15, line 15: '*The selected large attenuation of the channels prohibited an optimum detection of high-level dust layers and ice clouds.*'

Could you explain how the large attenuation prohibited an optimum detection of dust layers but allowed the determination of $\xi tot$ using the molecular depolarization ratio?

Page 15, line 15:

Typo: double space 'can be__noted'.

Page 17, line 2: *'based on three telescopes with a polarization filter on the front'.*

Do authors mean three 'channels'?

Page 17, line 13: *'However, it needs a strong depolarizing medium for its application, e.g., water*

*clouds.'*

This phrase might be confusing. Please, clarify that the strong depolarization comes from the multiple scattering not because the liquid droplets.

---

## Author Comment (AC1) · 24 Jan 2019

On behalf of the authors of the manuscript amt-2018-370, I would like to thank the reviewers for the feedback given during the first revision. We kindly invite you to have a look into the response document that we have prepared in order to answer the comments of the three reviews.

Please also note the supplement to this comment:
https://www.atmos-meas-tech-discuss.net/amt-2018-370/amt-2018-370-AC1-supplement.pdf
* * *
[Figure]

**Supplement:**

**Cristofer Jimenez, correspondence author from amt-2018-370 (cristofer.jimenez@tropos.de)**

**Response to reviews RC1, RC2 and RC3:**

On behalf of the authors of the manuscript, I would like to thank the anonymous referees for the constructive feedback given during the first revision of the manuscript. The suggestions and concerns were stated in a clearly and friendly manner, and have been very helpful to reveal writing errors and make the manuscript more comprehensible for readers. Each referee have also expressed concerns and suggestions regarding the presentation of the formalism and results and also about the discussion provided in the manuscript. The comments have been taken carefully into consideration and we have prepared an Answer document in order to address the specific comments of the three reviews RC1, RC2 and RC3.

We kindly invite the editor and reviewers to have a look into the corrected version of the manuscript, which is presented below this answer document. The changes done in response to the reviews have been highlighted with color in the corrected version of the manuscript. Three different colors were used, one for each review: RC1, RC2 and RC3.

**Answer to specific comments of RC1:**

1. Page 3, Line 9: "…consists of 2''achromatic lens…". Consider providing in a parenthesis also the equivalent of the 2'' in units of mm. **DONE**

2. Page 3, Line 13: Is the value of 650 m theoretically calculated or experimentally measured? In any case consider providing a reference at this point.
   **RC1.2 The full overlap altitude was estimated theoretically, the corresponding reference was already on the reference list but not cited on the text. This has been corrected.**

3. Page 5, Line 26: Maybe "described" is more appropriate than "considered".
   **RC1.3: Yes, I agree with this… DONE**

4. Figure 3: Consider annotating this figure with the letters (a), (b), (c), (d), in accordance to the figure legend. Moreover, make clear also in the figure that (a) and (b) refer to the emission while (c) and (d) to the receiving units. **DONE**

5. Page 7, Line 13: "…for the calibration is to insert an additional polarization…" **DONE**

6. Page 7, Line 21: "…, Müller matrices representing…" **DONE**

7. Page 7, Line 27: "$P_o$ is the number of emitted laser photons…" **DONE**

8. Page 8, Line 6: Consider replacing the sub scripts with capital characters, in order to be consistent with the annotation followed in the manuscript. **DONE**

9. Page 9, Line 12: "…, but depends on the receiver…" **DONE**

10. Page 10, Line 14: Eq. 38 is the product of Eq. 25 divided by Eq. 24, and not the inverse. **DONE**

11. In Eq. 42 I am missing the information of the variable C. Please specify to which quantity C refers to.

    **RC1.11** The interchannel constant $X_\delta$ use to be denoted by C. This equation was unintentionally skipped during the change of name from C to $X_\delta$. The authors apologize for this mistake that may have generated confusion during the 1st revision.

12. Page 11, Line 20: "… provides an overview of …" **DONE**

13. Page 11, Line 21: "(above ground level)" **DONE**

14. Consider presenting clearer the x-label of Figure 6 (e.g. $\eta_{tot} / [\eta_{\parallel,P} \cdot (1 + \varepsilon_r)]$) **DONE**

15. I would kindly suggest to the authors to show the profiles presented in Figure 7 up to higher altitudes (e.g. 4 km). Moreover, the profiles obtained by the ratio $N_P / N_{tot}$, compared to the rest two, seems to demonstrate greater variability with atmospheric height, in a way that I would say artificial layers are introduced. This can be seen for atmospheric heights inside the water cloud but also below (2.4 - 2.7 km). Is this also a result of the low SNR, even though that the profiles refer to 3 hours of measurement period? In any case the authors are kindly requested to comment on this.

    **RC1.15:** A second figure has been prepared for the corrected version of the manuscript. As this comment suggests, the profiles obtained with the ratio $N_P / N_{tot}$ present large variability on the low SNR region, which in our measurements seems more notorious, considering the large attenuation of the 3 channels (to avoid detector saturation at low level clouds). The points considered for the calibrations are however the altitudes where the depolarization is changing, which in clouds is also where the signal strength is large enough for been masked by signal noise. We have commented on this topic in the corrected version of the manuscript.

[Figure]

16. Figure 9 is very important. Legend: My suggestion is to use the phrase "(extended 3-signal method)". Additionally, it would be beneficiary for the manuscript if in the same figure, the profile of volume depolarization obtained by the conventional 3-signal technique (Reichard et al., 2003), is also shown. This will clearly demonstrate the improvement achieved by following the extended method proposed here, which takes into account various types of instrumental effects (e.g. the not perfectly polarized emitted laser light).

**RC1.16:** **One of the motivation to develop a new 3-signal approach was that the so called efficiencies ratios $D_i$ ($i = P, S, tot$) has to be known to apply the conventional 3-signal calibration (Reichardt et al., 2003). In the extended 3-signal approach this constants remain unknown. Later on the formulation they are combined with the effect of the angular misalignment between emitter and receiver and with the elliptically polarized laser beam into the global constant $\xi_{tot}$ (in the case of a quasy-ideal system, our case) or into $\xi_P$ and $\xi_S$ (for a non-ideal system).**

**In principle the 3-signal calibration approach from Reichardt et al. could be implemented, but only after applying our extended calibration approach and knowing the value of $\xi_{tot}$ (required for the calibration). We could add this profile on the Figure but it won't be a rigorous application of the conventional 3-signal calibration approach. To support the validation of the system and calibration approach, instead we added a second measurement for comparison (please see response RC2.14).**

17. The manuscript contains many equations and variables and this may easily confuse a reader. Therefore, I would kindly suggest to the authors to list all the variables used in a table (Appendix section), along with a small phrase describing them.

**RC1.17: Thanks for this suggestion. A list of variables has been added on the corrected version of the manuscript as Appendix B.**

**Answer to specific comments of RC2:**

1. Abstract: "A comparison with another polarization lidar" replaced by "A comparison with a second polarization lidar" **DONE**

2. Page 2, Line22-25: Please use the same notation for the sections. Currently we can find "section", "Section", "Sect." **DONE**

3. Section 3.1 starts with the statement that the manuscript follows the "notation and explanations of Freudenthaler (2016)" from AMT. Still in the following description, the authors define the misalignment between the polarization axis of the transmitted light and the copolarized receiver channel as θ - Page 5, Line 25-26: "The misalignment between the polarization axis of the transmitted light and the co-polarized receiver channel (defined by the respective polarization filter in front of the PMT) is characterized by angle θ ...."

The corresponding parameter in Freudenthaler (2016) should be the "Rotation of the plane of horizontal linear polarisation of the laser around the z axis (laser rotation)" which is relative to the receiving unit reference plane. Since the manuscript refers to a similar study performed by a lidar station from the same research network EARLINET-ACTRIS, I would suggest the authors to use the same notation used in the previous work (α). Keeping the same variable names and notations as used in previous studies will help a

reader familiar with similar studies and encourage the use of standardized variables and parameters.

4. Same comment as above applies for Page 6, Line 17: "The rotated polarization axis is represented in Fig. 3c, and after" and Figure 3 (also Page 8, Line 18).

**RC2.3-4**: **This sentence can lead to confusion indeed, since in our approach we do not adopt the whole nomenclature used in the mentioned recent studies (Freudenthaler 2016, Bravo-Aranda et al. 2016, Belegante et al. 2018).**

**The usage of $\alpha$ to describe the rotation angle was considered initially, but finally we opted for $\theta$, since the greek letter $\alpha$ may be confused with the atmospheric extinction coefficient (commonly represented by $\alpha$). The extinction is not included expressly on the equations since it does not play any role on the depolarization retrieval, some equations include however the term $\beta$, to describe the backscattering coefficient, which would have a completely different physical meaning than $\alpha$ representing the angle. Nevertheless, for the corrected version we have changed $\theta$ to $\alpha$ with the notation used in (Freudenthaler 2016) for the cosinus, i.e. $c_{2\alpha} = \cos(2\alpha)$.**

5. Page 7, Line 13: "A commonly used method for the calibration is  to insert"  **DONE**

6. Page 7, Line 27: "$P0$ is the  number of emitted laser photons"  **DONE**

7. Page 9, Line 25: This section should be described in more detail and the reasoning behind the use of two altitude heights should be clearly mentioned. Please consider extending this section since it is an important part of the theoretical background required to use the calibration technique used in this study.

**RC2.7: (please see RC2.7 & 9)**

8. Page 10, Line 3: "we obtained a mean value for $X_{\mathrm{p}}$". Is this really "p" or is this "δ"? **DONE**

**RC2.8: Yes, it should be $\delta$**

9. Page 10, Line 3-9: "Similarly, evaluation of many values of ..... are used to simultaneously determine the volume depolarization ratio in three different ways.". This section should be described in much more detail. Even if most of the readers are experts in lidar techniques, they are not familiar with the theoretical description and formalism presented in the manuscript. The theory behind this calibration technique is really

valuable since this is one of the few manuscripts dealing with the three channel calibration topic and it is important to provide a complete set of information on the theory. This section must be reconsidered before the manuscript is send for publication. **DONE**

**RC2.7 & 9:** **These comments have been taken carefully into consideration, since they reveal important weak points in the description of the method. The content in Page9, line 25 to page 10, line 8 has been reformulated in the corrected version of the manuscript.**
**The proposed change to the paragraphs from Page 10, line 3-9 is:**
*"In the conventional 3-signal calibration approach, each signal is normalized to a reference altitude, by doing so the efficiencies of the three channels $\eta_{II,P}$, $\eta_{\perp,S}$ and $\eta_{tot}$ cancel themselves from the equations, then the ratios between the three normalized signals are calculated. The retrieval of the volume depolarization ratio is done by solving a system of two equations and two unknowns: the volume depolarization ratio at a reference height $\delta(z_0)$ and the volume depolarization ratio at all heights $\delta(z)$ (Reichardt et al., 2003).*
*In this extended 3-signal calibration procedure, the signals are not normalized to a reference height $z_0$, instead, we divide directly the signals, obtaining the ratios $R_P$, $R_S$ and $R_\delta$, by taking then the difference between two altitudes (and not the ratio) we subtract the crosstalk in the emission and reception ($\varepsilon_l$ and $\varepsilon_r$) and the angular misalignment ($c_{2\alpha}$). The difference offers additionally a better performance in terms of error propagation compared to the ratio. In this way, the so called interchannel constants ($X_\delta$, $X_S$ and $X_P$) remain in the equations and they can be estimated by evaluating Eqs. (35), (36) and (37) respectively. Although we can estimate this three constants, we have to note that the number of unknowns are actually two $X_P$ and $X_S$ being the third constant $X_\delta$ the ratio of them (please see Eq. (30)), i.e. Eq.(35) is equivalent to Eq.(36) divided Eq. (37).*

*Given the form of Eqs. (35)-(37), observable differences between the height points $z_j$ and $z_k$ are needed for its evaluation, in practice, only altitude regions should be selected in the determination of $X_P$, $X_S$, and $X_\delta$ where significant changes in the depolarization ratio occur, e.g., in liquid-water clouds where multiple scattering by droplets produce steadily increasing depolarization with increasing penetration of laser light into the cloud (Donovan et al., 2015; Jimenez et al., 2017; Jimenez et al., 2018). Long measurements periods should be considered for the evaluation of Eqs. (35)-(37). All pair of data points ($z_j$ and $z_k$ in a certain height range, defined according to the ratio of signals) in all single measurements (in time t) provide an array with many observations of the interchannel constants, averaging these arrays we obtain a trustworthy estimate of these constants for the retrieval of the volume depolarization ratio (please see Figure 6)."*

10. Page 10, Line 14: "To derive  the linear depolarization" **DONE**

11. Page 10, Line 28: "In the first step, the inter-channel constant $X_\delta$ has to be measured." More detail must be provided by the authors. The experimental technique on how to perform the assessment of $X_\delta$ must be provided since this is one of the key parameters for the calibration of the depolarization channels.

    **RC2.11:** **We have modified this line in order to give more detail about the technique, since the constant $X_\delta$ has to be retrieved, not be measured directly.  As change for Page 10, line 8 we propose:**
    *"As first step of the calibration, the inter-channel constant $X_\delta$ (together with $X_P$ and $X_S$) is obtained from the measurements by evaluating Eqs. (35)-(37) in the selected height range (with variations on the depolarization) at each measurement time t."*

12. Eq. 42: please give more details on the missing variable "C".

    **RC2.12 (same as RC1.11):** **The interchannel constant $X_\delta$ use to be denoted by C. This equation was unintentionally skipped during the change of name from C to $X_\delta$.  The authors apologize for this mistake that may have generated confusion during the 1st revision.**

13. Page 13, Line 8: "By using constant $X_\delta$ and Eq. (42), a mean value of ..." Please provide more information on this topic.

    **RC2.13:** To provide more information we propose to change the sentence to:
    *Using the constant $X_\delta$ and evaluating Eq. (42) in the particle-free region of the 3-hour measurement period, a mean value of $\xi_{tot} = 1.118 \pm 0.008$ for the total crosstalk was obtained.*

14. Page 14, Line 10-15: Since the comparison between the volume linear depolarization ratio measured by MARTHA and BERTHA is designed to validate the calibration technique used in this study, I would advise to also use a second case for this comparison. A strong depolarizing layer (e.g mineral) would help validate the results for highly depolarizing layers.

    **RC2.14:** To support the comparative validation. A second simultaneous measurement case was considered for comparison. Page 14 lines 10-15 and Figure 9 have been updated as follows:

*To validate the new system and the calibration procedure a comparison between the measurements of the volume linear depolarization ratio with the lidar systems MARTHA and BERTHA (Backscatter Extinction Lidar Ratio Temperature and Humidity profiling Apparatus) is presented in Figure 9. The observations were conducted at Leipzig (51°N, 12°E) on 29 May 2017 with the presence of a Dust layer between 2 and 5 km and a cirrus cloud at 11 km (see Fig. 9a). Good agreement in the dust layer can be noted, while the cirrus cloud shows differences between the two systems, that difference can be attributed to the fact that the BERTHA system is pointing 5° respect to the zenith, while the MARTHA system points to the Zenith (0°). This could lead to specular reflection by horizontally oriented ice crystals reducing the depolarization ratio in the case of the MARTHA system.*

*A second measurement period during an unique event with a dense biomass burning smoke layer in the stratosphere on 22 August 2017 was considered for comparison (Haarig et al., 2018), here very good agreement for the layer between 5 and 7 km and also for the layer at 14 km was obtained, confirming the good performance of the systems and of the respective calibration procedures, extended 3-signal method in MARTHA and the Δ90° method in the BERTHA system.*

[Figure]

**Figure 9: Volume linear depolarization ratio obtained with MARTHA (extended 3-signal method) and BERTHA (Δ90° method) on (a) 29 May 2017 20:20-20:45 (with smooth 27 bins) (b) 22 August 2017 20:45-23:15 (Haarig et al., 2018).** The system were located at a distance of 80 meters and were calibrated independently.

15. Page 17, Line 22-25: I do not see the necessity of this section. A link with further studies was already included in the introduction of the study and since this section is

not connected to the conclusions I would advise to remove it for the final version of the manuscript. **DONE**

16. Since the manuscript has an important theoretical section containing many variables and equations, I would suggest adding an additional list of variables containing a comprehensive description for each element. Please consider following the same terminology used by Freudenthaler (2016) .

**RC2.16: Thanks for this suggestion. A list of variables has been added on the corrected version of the manuscript as Appendix B.**

**Answer to specific comments of RC3**

1. Page 4, line 3: *'In the alignment process, the cross-polarized axis is found when the count rates are at the minimum.'*

This process seems to be inaccurate because the change of the signal due to several degrees can be masked by the signal noise. Did you check the accuracy of this procedure? Could you provide the uncertainty?

**RC3.1: To find the minimum in the channels P and S we reduce their attenuation (Figure 2) in order to increase the signal strength as much as possible and so avoid noisy signals. The process is still a little bit rudimentary, since the minimum is found by eye rotating manually the mounted filter on the top of the telescopes P and S, so the uncertainty inherent to this aligning process cannot be reported. Therefore, the angular misalignment of the P and S channels with their respective component axis $II\ and \perp$ was considered on a first stage as unknown. After applying the new 3-signal calibration procedure, the overall impact on the channels P and S of this angular misalignments, added to the crosstalks of the emission and reception units ( $\xi_{tot}$ or $\xi_P$ and $\xi_S$) can be estimated.**

2.  Page 4, line 18: *'Based on this theoretical framework we will derive three lidar equations for our three measured signal components.'*

I think that the theoretical framework has not been presented yet. Please, considered to change by 'Based on the theoretical framework of ___, we will derive [...]'.

**RC3.2**:  **Indeed. We propose the change for Page 4, line 18**:
*"As first step in this theoretical framework, we will derive…"*

3.  Page 5, lines 10-14: *'In our approach, …'*

I recommend to mention the Figure 3 somehow. It will be easier to understand this paragraph following the steps of Figure 3.

**RC3.3: Thanks for this suggestion. An indication of the figure has been added in parenthesis on the manuscript.**

4.  Page 5, line 18: *'We introduce the so-called crosstalk term $\varepsilon_i$'*

In the line 1 of the same page, it is stated that the notation and explanations of previous manuscripts are used. However, the term 'cross-talk' is mainly used to describe non-ideal beamsplitter cubes instead of the depolarization of the outgoing laser light (emitting block). If I properly understood, the cross-talk term would correspond to the depolarisation of the laser light after crossing the transmission block (a king of linear polarisation parameter $a_L$, according to the Freudenthaler's paper). Additionally, I would say that the angle $\theta$ in this manuscript corresponds to the angle $\alpha$ in the Freudenthaler's paper. For the sake of clarity, it would be very helpful for the community whether the same nomenclature is used or, at least, a small mention about the connections is included.

**RC3.4**: **I agree that this sentence can lead to confusion, since in our approach we do not adopt the whole nomenclature used in the mentioned recent studies (Freudenthaler 2016, Bravo-Aranda et al. 2016, Belegante et al. 2018).**
**The usage of $\alpha$ to describe the rotation angle was considered initially, but finally we opted for $\theta$, since the greek letter $\alpha$ may be confused with the atmospheric extinction coefficient (commonly represented by $\alpha$). The extinction is not included expressly on the equations since it does not play any role on the depolarization retrieval, some equations include however the term $\beta$ to describe the backscattering coefficient, which would have a completely different**

physical meaning than $\alpha$ representing the angle. Nevertheless, for the corrected version we changed $\theta$ to $\alpha$ with the notation used in (Freudenthaler 2016) for the cosines, i.e. $c_{2\alpha} = \cos(2\alpha)$.

The crosstalk of the emission $\varepsilon_l$ indicates the depolarization of the light after the transmission block, which would be similar to the linear polarization parameter $a_l$, adopted in Freudenthaler (2016), but in terms of depolarization.

In the manuscript the term crosstalk is intended to describe the undesirable component on the respective polarization axis, which is assumed to be present during the emission and reception of the wave fronts. In this sense, crosstalk would also hold for the non-ideal response of polarizing beamsplitters.

5. Page 6, line 2 and Eq (6):

According to the Freudenthaler's paper, the rotation R(θ) proposed in this manuscript is a particular case of the emitting block, being other polarizing effects omitted such as the diattenuation. It would be helpful for the readers to have a list of parameters considered ideals.

**RC3.5:** **In the approach further polarizing effects, such as diattenuation and retardation are assumed as ideals, in the emission, characterized by the elliptical polarized wave-front, and also in the reception, where the polarization state of the light is filtered at the beginning of the optical path. Now in Appendix B, indication the parameters assumed as ideals, can be found.**

6. Page 6, Figure 3:

It would be helpful for future readers to link each step in the figure with the term, as follow:

[Figure]

**RC3.6: Thanks for this suggestion. It has been implemented on the manuscript. The scheme is much more illustrative now.**

7. Page 7, line 13: *'A commonly used method for the calibration is the to insert an extra polarization filter[…]'* **DONE**

Typo? **RC 3.7: Yes**

8. Page 8, line 6: *'Because identical polarization filters are used in our lidar setup, we can assume $Dp = Ds^{-1}$.'*

What do you mean with 'identical'? Even the same model of polarizer made by the same company might show quite different behaviors. Additional details should be addressed to support this sentence.

**RC3.8:  With identical we meant the same filter model. Although the filters may present different extinction ratio, in our approach the difference between $D_P$ and $D_S^{-1}$ are considered as neglectable, since their value should be less than $10^{-3}$ for 532 nm (according to the fabricant).**

**Considering the mean value of total crosstalk obtained from the measurements of 2017 ($\xi_{tot} = 1.109 \pm 0.009$), it seems that the parameters that really have an impact on the system (in terms of polarization) are the rotation of the polarization plane $\theta$ (now $\alpha$) and the crosstalk from the emitted laser ($\varepsilon_l$), given the form of this parameter (Eq. (39) on the manuscript).**

$$\xi_{tot} = \frac{(1+\varepsilon_r)(1+\varepsilon_l)}{(1-\varepsilon_l)(1-\varepsilon_r)\cos(2\theta)} \geq 1, \tag{39}$$

**At the end, the manuscript propose two calibration methods. 1) For a quasy-ideal system ($D_P = D_S^{-1}$ and $\theta_P = \theta_S$, our case) and for a non-ideal system ($D_P \neq D_S^{-1}$ and $\theta_P \neq \theta_S$) (outlined in Appendix A). Lidar users should estimate which approach represent more accurately their system.**

**We propose to modify Page 8, line 6 to:**

*"The absence of optical elements before the polarization filters (such as the telescope and beamsplitters) avoids further polarization effects, such as diattenuation and retardation (Freudenthaler, 2016). Moreover, since we employed the same filter model in the optical path of the channels P and S, we assumed that $D_P = D_S^{-1}$."*

9. Page 9, line 2: ' *[…] based on a measurement example, we demonstrated that the impact of this assumption can be neglected in our system.'*

Could you provide any indication about the validity of this assumption in other systems?

**RC 3.9:** **Up to now, the only system in our facilities with the three signal implemented is the MARTHA system, therefore the validity of this assumption can only be supported by the results obtained with it.**

10. Page 10, line 11: *'liquid-water clouds where multiple scattering by droplets produce steadily increasing depolarization with increasing penetration of laser light into the cloud […]'* and Page 17, line 21: *'the volume of the depolarization ratio does not depend on the field of view of the receiver, however in multiple scattering regime (e.g. in liquid water clouds), it does […]'.*

Could the multiple scattering be a problem for the depolarization calibration?

**RC 3.10:** **The multiple scattering on water droplets is what produce depolarization and does not represent a problem to the calibration (excepting the fact that multiple scattering may increase the signal strength to saturation levels, if the attenuation of the channels is not large enough).**

**The first sentence (Page 10, line 11) aims to indicate that the profile of depolarization in the cloud offers a wide range of values to retrieve the interchannel constants (based on the difference of signal ratios among different heights). The second sentence (Page 17, line 21) is written to indicate that the dependence on the FOV size permits us to assess cloud microphysics by means of depolarization at two or more FOVs. As suggested by the 2nd Review (RC2.15), this sentence will be removed since the link with further studies is already included in the introduction.**

11. Page 10, line 29: 'Then $\xi tot$ can be estimated in a region (defined by height zmol) with dominating Rayleigh backscattering […]'

This the most important handicap I detect in this method. $\xi tot$ must be estimated in a particlefree region where the SNR used to be quite low. This is the same handicap of the classical molecular calibration, including a more complicate ldiar system since three channels are required instead of two. So, why is this method more advisable?

The main difference between with the classical molecular calibration method is that in this approach the molecular region is used to estimate the so called total crosstalk parameter ($\xi tot$).  This constant summarizes the impact of the considered systematic error sources (crosstalk in the emission ($\varepsilon_l$) and reception ($\varepsilon_r$) path and angular misalignment between emission and reception ($\cos(2\theta)$ now denoted as $c_{2\alpha}$), and it is expected to be constant with time.  On the other hand, the so called interchannel constants ($X_P, X_S$ and $X_\delta$) describe the ratio between the efficiencies of the 3 channels P,S and *tot* to their respective components ( ‖ , ⊥ and ‖ +⊥) , these efficiencies ($\eta_{II,P}, \eta_{\perp,S}$ *and* $\eta_{tot}$) represent the product of the effective area of the receiver, the transmissivity of the optical path (modulated by the attenuation setup) and also the gain and efficiency of the detectors, These constants ($X_P, X_S$ and $X_\delta$) are expected to vary with time, as the attenuation is eventually change by users, and also as the efficiency of the detectors decays with time.

This method would be more advisable since it separates the unknowns of the problem in constants that vary with time ($X_P, X_S$ and $X_\delta$)  and a constant that does not change with time ($\xi tot$) estimated in the particle-free region. When averaging long term measurements, we can get rid of the eventual bias induced by aerosol particles present in the region considered as free-particle region, allowing an accurate estimate of this non-changing constant.

12. Page 11, Eq. (42):

Which is the meaning of the term 'C'? I was not able to find its definition. From Eq. (41) to Eq.

(42), I got that $C = X_\delta$. Please, specify it.

**RC3.12 (same as RC1.11 & RC2.12):** The interchannel constant $X_\delta$ use to be denoted by C. This equation was unintentionally skipped during the change of name from C to $X_\delta$. The authors apologize for this mistake that may have generated confusion during the 1st revision.

13. Page 12, line 10 and caption of Figure 6:

Whereas it is stated that the height range goes from a few meters below the cloud base up to 240 m above (page 12 line 10) in the caption of the Figure 6, it is noted that 16000 data points were obtained. Could you explain the huge number of data points in this small height range?

**RC3.13:** For the calculations of the calibration constants, each pair of bins in the 240 meters are considered for the evaluation of Eqs. (35)-(37). In a 5 minutes profile we get one result at each combination of height bins in the selected range, for 32 height bins (240 meters), we get $\sum_{n=1}^{31} n = 496$ combinations. Considering all the 5-min profiles in the hours of measurement we get actually 17856 data points. The Figure label has been corrected to the right amount of points (about 18000).

14. Page 13, line 8: *'The crosstalk factor has a large impact on the retrieval of the volume linear depolarization ratio only in the region with low depolarization ratios.'*

Please, include whatever is necessary to demonstrate this sentence.

**RC3.14:** We have reformulated this sentence to make clearer the point. We changed the Page 13, line 8 to:

*"Given the form of the equations to retrieve the profiles of volume depolarization ratio (Eqs. (35)-(37)), the propagated uncertainty associated to $\xi_{tot}$ does not vary largely with height, which leads to a large percentage uncertainty on the retrieval of the volume linear depolarization ratio in the region with low depolarization ratios, also characterized by low signal strengths"*

15. Page 15, line 8: *'4.2 Long-term stability of the polarization lidar calibration and performance'* and Page 17, line 18: *'Long term studies indicated the robustness and stability of the three-signal lidar system over long time periods.'*

The calibration stability was analyzed between April and November 2017 (8 months). I would use 'long-term' for larger periods and thus, I suggest replacing 'long-term stability' by 'temporal stability'. **DONE**

16. Page 15, line 15: *'The selected large attenuation of the channels prohibited an optimum detection of high-level dust layers and ice clouds.'*

Could you explain how the large attenuation prohibited an optimum detection of dust layers but allowed the determination of $\xi tot$ using the molecular depolarization ratio?

**RC3.16:** **It is meant that on this layer, the determination of the interchannel constants is not optimal, since it requires the differences of the signal ratios between single height bins. The large attenuation on the channels for this system (since this system particular aims to measure depolarization in liquid clouds) reduce the SNR in aerosol layers making difficult to use the mentioned difference of ratios. For the estimation of $\xi tot$, we can average the whole measurement period and a large height range reducing considerably the impact of the noise of the measurements.**

**We have changed this sentence to:**

*"One reason for these differences in the uncertainty of $X_\delta$ is that the system was optimized for the observation of low-altitude liquid-water clouds, for which the detection channels need large attenuation to avoid saturation of the detectors in the cloud layer. This setup prohibited an optimum detection of high-level dust layers and ice clouds due to the low signal strength for these cases."*

17. Page 15, line 15:

Typo: double space 'can be__noted'. **DONE**

18. Page 17, line 2: *'based on three telescopes with a polarization filter on the front'*.

Do authors mean three 'channels'?

**RC3.18:** **We meant telescopes indeed. We propose the change on the text:** *"based on three telescopes (one for each channel) with a polarization filter on the front"*

19. Page 17, line 13: *'However, it needs a strong depolarizing medium for its application, e.g.,*

    *water clouds.'*

This phrase might be confusing. Please, clarify that the strong depolarization comes from the multiple scattering not because the liquid droplets.

**RC3.19:** **This can be confusing indeed. To avoid confusion with this sentence, we propose to change it to:**

[revised manuscript text omitted]